# Oxidation of Methanol and Dichloromethane on TiO_2_-CeO_2_-CuO, TiO_2_-CeO_2_ and TiO_2_-CuO@VUKOPOR^®^A Ceramic Foams

**DOI:** 10.3390/nano13071148

**Published:** 2023-03-23

**Authors:** Lenka Matějová, Ivana Troppová, Satu Pitkäaho, Kateřina Pacultová, Dagmar Fridrichová, Ondřej Kania, Riitta Keiski

**Affiliations:** 1Institute of Environmental Technology, CEET, VSB-Technical University of Ostrava, 17. Listopadu 2172/15, 708 00 Ostrava-Poruba, Czech Republic; 2Environmental and Chemical Engineering, Faculty of Technology, University of Oulu, P.O. Box 4300, FI-90014 Oulu, Finland; 3Elvac Ekotechnika s.r.o, Tavičská 337/23, 703 00 Ostrava-Vítkovice, Czech Republic

**Keywords:** catalytic oxidation, volatile organic compounds, methanol, dichloromethane, titania, ceria, copper oxide, mixed oxide catalyst, VUKOPOR^®^A ceramic foam

## Abstract

The application-attractive form of TiO_2_, CeO_2_ and CuO-based open-cell foam supported catalysts was designed to investigate their catalytic performance in oxidation of two model volatile organic compounds—methanol and dichloromethane. TiO_2_-CeO_2_, TiO_2_-CuO and TiO_2_-CeO_2_-CuO catalysts as thin films were deposited on VUKOPOR^®^A ceramic foam using a reverse micelles-controlled sol-gel method, dip-coating and calcination. Three prepared catalytic foams were investigated via light-off tests in methanol and dichloromethane oxidation in the temperature range of 45–400 °C and 100–500 °C, respectively, at GHSV of 11, 600 h^−1^, which fits to semi-pilot/industrial conditions. TiO_2_-CuO@VUKOPOR^®^A foam showed the best catalytic activity and CO_2_ yield in methanol oxidation due to its low weak Lewis acidity, high weak basicity and easily reducible CuO species and proved good catalytic stability within 20 h test. TiO_2_-CeO_2_-CuO@VUKOPOR^®^A foam was the best in dichloromethane oxidation. Despite of its lower catalytic activity compared to TiO_2_-CeO_2_@VUKOPOR^®^A foam, its highly-reducible -O-Cu-Ce-O- active surface sites led to the highest CO_2_ yield and the highest weak Lewis acidity contributed to the highest HCl yield. This foam also showed the lowest amount of chlorine deposits.

## 1. Introduction

Emissions of volatile organic compounds (VOCs) into air and water from products or processes is considered as serious environmental problem [1,2]. According to the classification of the International Agency for Research on Cancer (IARC classification) some of widely used VOCs belong to the three most serious groups; group 1 (carcinogens, e.g., formaldehyde, vinyl chloride), group 2A (probable carcinogens to humans, e.g., halogenated VOCs such as dichloromethane, perchloroethylene) and group 2B (possible carcinogens to humans, e.g., acetaldehyde, halogenated VOCs such as chloroform) [3]. VOCs such as alcohols (e.g., methanol, 2B group) are toxic to human health and cause serious health problems such as blindness or chronical kidney and liver diseases. Besides primary emissions, emitted VOCs can react with other gases and particulates to form secondary air pollutants that in some cases are even more harmful than the original VOC emissions complicating e.g., their abatement.

From all VOCs, methanol and dichloromethane belong to solvents widely used in pharmaceutical and chemical industry for chemical synthesis or laboratory analytical analyses, being also part of products such disinfectants, degreasers, varnishes, waxes, household cleaners, used in production of paints and furniture materials [2]. Handling both of them, disposal of used solvents, and the chosen emission abatement technology must be safe and under strict control since even more harmful by-products such as formaldehyde or carbon monoxide may be produced and emitted. Depending on their categorisation, VOCs are regulated by a wide variety of governmental agencies [2]. Catalytic combustion is a well-accepted and reliable technology for industrial VOCs abatement in environmentally and economically acceptable manner [2,4].

From a practical point of view, there are given strict requirements on the catalysts applied for VOCs catalytic combustion, i.e., high activity, high selectivity as well as good durability. Concerning catalytic oxidation of a broad range of individual VOCs over individual types of catalysts some review papers were already reported [1,2,4,5,6], also including investigated confined-structured catalysts [6] and by-products analysis for chlorinated VOCs (CVOCs) oxidation [7].The main challenges which need to be still further solved concerning the development of catalysts for VOCs oxidation are the development of highly selective and durable catalysts, deeper knowledge and prediction of oxidation mechanisms [4,5], catalyst deactivation, development of more complex catalytic systems being able to oxidize a range of VOCs at much lower temperatures [1] and with respect to character of contaminated sites the coupling process combining different VOCs/CVOCs abatement methods should be applied to enhance the removal efficiency in a more cost effective way [1,2,5]. Concerning the confined-structured catalysts more easy, available and green synthetic methods should be developed and used, including the optimization of chemical composition. There also arise some limitations in their precise physicochemical characterization, which is necessary for understanding of their catalytic performance. There is also a lack of testing of confined-structured catalysts in more realistic working conditions and oxidation of a mixture of VOCs [6]. For CVOCs oxidation, (i) the examination of Cl migration and evolution pathway via advanced modelling and characterization techniques, providing the information how to prevent unwanted chlorinated by-products formation with working adjustment, (ii) revelation of proper metal combination within catalyst design, (iii) definition of the exact role of each physicochemical property in deep oxidation and chlorination steps with the aim to tailor catalyst surface composition, inhibiting the formation of unwanted chlorinated by-products and (iv) enhancing oxidation efficiency at low temperature that can evade Cl_2_ formation and chlorination reactions, are of keen interests. It was suggested that e.g., technological coupling of hydrodechlorination, catalytic oxidation and ozone-assisted oxidation is feasible approach how to enhance total elimination at low temperature without chlorination reactions. (v) Last but not least, investigation of the effect of other additives and oxidizing agents, promoting dechlorination reactions and inducing Cl species into final products, is also necessary [7]. Based on above mentioned facts the development of industrially-attractive and robust form of catalyst based on mixed metal oxides, which are cheaper and more resistant against metal chlorination than noble metal-based catalysts, using simple chemical and deposition route, is the logical motivation for the research. Moreover, the developed form of catalysts should be examined in oxidation of more model VOCs and in the presence of water to reveal the catalyst selectivity and be closer to realistic adjustment, yielding more desired HCl than Cl_2_. 

The TiO_2_-CeO_2_-CuO ternary mixture is attractive oxidation catalyst in terms of good temperature stability, adjustable acid-base properties, easy reducibility and the ability to store oxygen in its crystal structure [8]. CuO-based materials are suitable catalysts for environmental remediation, but their disadvantage is easy sintering and deactivation due to blockage of active sites e.g., by carbon deposits and the associated reduced catalyst stability [9]. Cerium, respectively, CeO_2_, serves as a suitable CuO promoter due to the promotion of redox properties (there can be an interaction between the redox pairs Cu^2+^/Cu^0^ and Ce^3+^/Ce^4+^) and oxygen storage capacity due to the presence of Ce^3+^/Ce^4+^ [10,11]. Furthermore, the presence of CeO_2_ helps the thermal stability of the oxide mixture and reduces the formation of coke on its surface [12]. Nano CeO_2_ and Cu-doped nano CeO_2_ are even effective in the oxidation of particulate matter/soot [13,14]. The CeO_2_ and CuO also interacts strongly with TiO_2_ support [15]. Ce doped TiO_2_ increases Cu activity and stabilizes the Cu/TiO_2_ surface [15]. Concerning TiO_2_-CeO_2_ mixtures, the addition of cerium/CeO_2_ to TiO_2_ decreases the TiO_2_ anatase crystallite-size and increases its thermal stability [16,17], increases reducibility [18] as well as Lewis and Brønsted acidity [19].

Concerning methanol oxidation (oxygenated VOC), the existing literature reveals that TiO_2_, CeO_2_ and CuO-based catalysts are effective catalysts in the decomposition of methanol to H_2_, respectively to a mixture of H_2_ and CO (so-called syngas) which can be an alternative fuel [11,20,21], or have been studied in the oxidation of ethanol, ethylacetate and CO [11,15,22,23] (Table 1). These catalysts have been rarely studied in the deep methanol oxidation, where the required reaction products are CO_2_ and H_2_O [23,24], whereas being cheaper and more efficient equivalent to catalysts doped with noble metals.

Concerning dichloromethane oxidation (chlorinated VOC, CVOC), a few interesting studies indicating good potential of TiO_2_, CeO_2_ and CuO-based catalysts instead of more expensive noble metals-based catalysts were reported [18,25,26,27,28] (Table 1). However, published research from Cao et al. [18,25] announced interestingly some existing limitations related to the design of the mixed TiO_2_-CeO_2_-CuO catalyst and its higher tendency to deactivation compared to a separate two step catalytic system Ce/TiO_2_-Cu/CeO_2_ applied in a series. Cao et al. studied the dichloromethane oxidation in dry conditions, firstly over TiO_2_ and Ce/TiO_2_ [18], followed by a study on the Ce/TiO_2_-Cu/CeO_2_ in a series [25]. Within their first study [18], they suggested a two-step mechanism of dichloromethane oxidation over dehydrated TiO_2_ and Ce/TiO_2_, explaining the production of unwanted by-products such as methylchloride (CH_3_Cl), formic acid (HCOOH) and carbon monoxide (CO) over TiO_2_. Ce/TiO_2_ showed significantly improved catalytic activity as well as selectivity to CO and CO_2_ in dichloromethane oxidation compared to TiO_2_. Moreover, Ce/TiO_2_ showed to be stable during the 100 h test. They proved the rapid removal of Cl on the surface of CeO_2_ (on oxygen vacancies of ceria), preventing TiO_2_ (Ti^4+^ sites) deactivation by adsorbed chlorine, accompanied by CeO_2_′s capability to enhance further deep oxidation of C–H from by-products and retain a certain oxidation of CO to CO_2_ [18]. In their second study Cao et al. [25] suggested a two-step catalytic system Ce/TiO_2_-Cu/CeO_2_ in a series to reach total oxidation of dichloromethane. The investigated catalytic system achieved improved activity as well as CO_2_ selectivity with less undesired CO, Cl_2_ and by-products (C_x_H_y_Cl_z_) in dry air. It showed good stability in long-term test in the presence of water. For the Ce/TiO_2_-Cu/CeO_2_ in a series they suggested a three-step dichloromethane degradation mechanism consisting of (i) dichloromethane adsorption and rupture of the C–Cl bonds, (ii) deep oxidation of C–H from by-products and (iii) total oxidation of CO to CO_2_. The Ce/TiO_2_-Cu/CeO_2_ system in a series ensures that the rupture of C–Cl and the total oxidation of CO are physically separated, avoiding the decrease in acid sites to happen on a (Ce + Cu)/TiO_2_ catalyst and the chlorine poisoning of TiO_2_ due to the strong adsorption of Cl on CuO. Moreover, CuO itself had a high resistance to the toxicity of Cl [25]. El Assal et al. [26] have examined the efficiency of V, Mn and Cu supported on TiO_2_, MgO and CeO_2_ in dichloromethane oxidation. The highest selectivity to HCl was reached by Cu/TiO_2_, followed by V_2_O_5_/MgO. Pure TiO_2_ support was the most acidic support from all investigated and its highest acidity correlated well with the best activity and selectivity in dichloromethane oxidation. Impregnation of Cu onto TiO_2_ increased the total catalyst acidity, resulting in better dichloromethane adsorption and cleavage of the C–Cl bond, leading to the best HCl selectivity, the least formation of by-products and the highest catalytic activity. Concerning the selectivity of the examined catalysts, both undesired chlorinated and oxygenated by-products such as methyl chloride (CH_3_Cl), chloroform (CHCl_3_), formaldehyde (CH_2_O) and carbon monoxide (CO) were identified in dichloromethane oxidation experiments. From this point of view Cu/TiO_2_ produced the lowest concentrations of the mentioned undesirable by-products [26]. Wang et al. [28] tested M/TiO_2_ and Pd-M/TiO_2,_ where M was Ce, V or Mn in dichloromethane oxidation. While Ce/TiO_2_ and Pd-Ce/TiO_2_ showed the best catalytic activity, Ce/TiO_2_ showed very poor selectivity to CO_2_. Pd modification of Ce/TiO_2_ led to the improvement of the catalyst redox ability and surface adsorbed oxygen species which was reflected to significant improvement of selectivity to CO_2_ of Pd-Ce/TiO_2_, but still with detected high concentrations of CO (100 ppm) at 100% dichlromethane conversion. Concerning unwanted chlorinated by-products, they detected traces of methyl chloride (CH_3_Cl) over the tested catalysts. The performed DFT calculations and in-situ DRIFT studies over the best Pd-V/TiO_2_ catalyst helped to reveal the mechanism of dichloromethane oxidation: C–Cl bonds and C–H bonds are dissociated by the surface lattice oxygen species after the adsorption of molecular dichloromethane on the acid sites, producing intermediates. After that, the oxidation of CO to CO_2_ takes place by the surface adsorbed oxygen species.

**Table 1 nanomaterials-13-01148-t001:** Overview of TiO_2_, CeO_2_ and CuO-based catalysts investigated in methanol oxidation/decomposition, dichloromethane oxidation or oxidation of their possible reaction by-products.

Catalyst	VOC/Reaction	Reference
20%CuO/CeO_2_	methanol oxidation	[24]
TiO_2_-CeO_2_	methanol decomposition ethylacetate oxidation	[11]
TiO_2_-CeO_2_-CuO	methanol decomposition ethylacetate oxidation	[20]
TiO_2_-CeO_2_-CuO	ethanol, ethylacetate and CO oxidation	[22]
5% CuO_x_-TiO_2_	CO oxidation	[15]
Pd, Pt-Pd/CeO_2_-Al_2_O_3_-TiO_2_	methanol oxidation	[23]
CuO-TiO_2_, CuO-CeO_2_	dichloromethane oxidation	[26]
TiO_2_, Ce/TiO_2_	dichloromethane oxidation	[18]
Ce/TiO_2_-Cu/CeO_2_	dichloromethane oxidation	[25]
Cu-Mn/Al_2_O_3_	ethanol and dichloromethane oxidation	[27]
M/TiO_2_, Pd-M/TiO_2_ (M-Ce, V, Mn)	dichloromethane oxidation	[28]

The aim of this study was to develop the application-attractive forms of TiO_2_, CeO_2_ and CuO-based open-cell foam supported catalysts and investigate their activity, selectivity and stability in oxidation of two model VOCs—dichloromethane (chlorinated VOC) and methanol (oxygenated VOC). Thus, binary TiO_2_-CeO_2_ and TiO_2_-CuO catalysts and ternary TiO_2_-CeO_2_-CuO catalyst as thin films deposited on VUKOPOR^®^A ceramic Al_2_O_3_-based foam were prepared by using a reverse micelles-controlled sol-gel method in combination with dip-coating and calcination. This preparation method is feasible in industrial scale due to high chemical stability of the sol from the reverse micelles-mediated sol-gel synthesis and leads to crystallization of metal oxides of small crystallite size, which does not change significantly with rising temperature [29,30], crystallite size is typically <18 nm, which may be positively reflected to their catalytic performance and stability. Using dip-coating method and calcination the catalytic thin film is very well-fixed on the foam due to its roughness and chemical bonding. These catalytic foams were prepared for testing in these two oxidation reactions for the first time. The activity and selectivity of 3 catalytic foams in dichloromethane and methanol oxidation were studied and compared based on light-off tests and the best catalytic foam was then exposed to 20 h stability test. Special attention was dedicated to studying the selectivity over the studied catalytic foams in both oxidation reactions since this issue is very important in catalyst development for VOC oxidation in industrial scale.

## 2. Materials and Methods

### 2.1. Preparation of TiO_2_, CeO_2_ and CuO-Supported VUKOPOR^®^A Ceramic Foams

The sol-gel method controlled in reverse micelles environment of nonionic surfactant Triton X-114 in cyclohexane combined with dip-coating and calcination was selected for preparation and deposition of TiO_2_-CeO_2_, TiO_2_-CeO_2_-CuO and TiO_2_-CuO catalysts (4 layers) on VUKOPOR^®^A ceramic foams (85% Al_2_O_3_ + 14% SiO_2_ +1% MgO, porosity 40 ppi, diameter × length = 8 × 20 mm, LANIK s.r.o., Boskovice, Czech Republic). Firstly, the sols were prepared by mixing cyclohexane, Triton X-114, water and Ti-Ce-Cu precursors dissolved in anhydrous ethanol. The metal precursors used for synthesis were Ce(NO_3_)_3_·6H_2_O, Cu(NO_3_)_2_·3H_2_O and titanium (IV) isopropoxide (Ti(OCH(CH_3_)_2_)_4_, purity < 97%) from Aldrich (St. Louis, MO, USA).

For the preparation of TiO_2_-CeO_2_-CuO-supported VUKOPOR^®^A ceramic foam, the synthesis was as follows. 3.26 g of Ce(NO_3_)_3_·6H_2_O and 0.55 g of Cu(NO_3_)_2_·3H_2_O were individually dissolved in 5 mL of anhydrous ethanol. After complete dissolution of both precursors these two solutions were mixed, and the beakers were rinsed additionally with 5 mL of ethanol. The CeCu-based solution was stirred for 10 min. Meanwhile, 91.9 mL of cyclohexane, 37.6 mL of Triton X-114 and 0.46 mL of water was mixed to form Triton-based sol, which was stirred for 20 min. Then, the CeCu-based solution was added to the Triton-based sol and the formed micellar sol was stirred for 15 min. After that, 20 mL of titanium (IV) isopropoxide was added dropwise and the final Ti/Ce/Cu-based micellar sol was mixed for further 20 min. The transparent homogeneous sol was left standing in closed flask for minimally 24 h. Then, the washed VUKOPOR^®^A ceramic foams were immersed into the Ti/Ce/Cu-based micellar sol using dip-coater iDLab, using the following conditions of dip-coating (immersion velocity—150 mm/min, delay in the sol—60 s, emergency velocity—60 mm/min). The deposited foams were left to dry on air for 4 h and then were calcined at 500 °C for 4 h with the heating rate 3 °C·min^−1^.

For the preparation of TiO_2_-CeO_2_-supported VUKOPOR^®^A ceramic foam, the synthesis steps were identical as in the case of TiO_2_-CeO_2_-CuO foams, but the used amounts of precursors were following: 89 mL of cyclohexane, 36.5 mL of Triton X-114, 0.54 mL of H_2_O, 3.26 g of Ce(NO_3_)_3_·6H_2_O and 20 mL of titanium (IV) isopropoxide.

For the preparation of TiO_2_-CuO-supported VUKOPOR^®^A ceramic foam, the synthesis steps were identical as in the case of both previous foams, but the used amounts of precursors were following: 83 mL of cyclohexane, 34 mL of Triton X-114, 1.136 mL of H_2_O, 0.547 g of Cu(NO_3_)_3_·3H_2_O and 20 mL of titanium (IV) isopropoxide.

The appearance of parent VUKOPOR^®^A ceramic foam and the prepared catalytic foams is shown in Figure 1.

### 2.2. Physicochemical Characterization of TiO_2_, CeO_2_ and CuO-Supported VUKOPOR^®^A Ceramic Foams

X-ray fluorescence spectrometry (XRF) was implemented in order to determine the real chemical composition of the catalytic foams. The mass content of TiO_2_, CeO_2_ and CuO (in wt.%) was determined semi-quantitatively. XRF measurements were made on a dispersion spectrometer XEPOS (Spectro, Kleve, Germany) in a protective atmosphere (He), using a powder produced from the crashed catalytic foam, processed into a tablet with a diameter of 24 mm.

Textural properties of the prepared catalytic foams (i.e., specific surface area, S_BET_, total pore volume, V_total_, mesopore and macropore-size distribution) were evaluated on the basis of measurements of physical nitrogen adsorption at 77 K. Physisorption measurements were carried out on the 3Flex instrument (Micromeritics, Norcross, GA, USA). Prior to the physisorption measurement, the foams (4 pieces, containing ~0.14 g of deposited oxidic catalyst) were degassed under vacuum (<1 Pa) for ~24 h at 105 °C to remove the physically adsorbed water. The dried foams were then subjected to nitrogen physisorption at 77 K. The specific surface area, S_BET_, was evaluated using the Brunauer-Emmett-Teller (BET) theory [31] for the range of relative pressures p/p_0_ = 0.05–0.30. The total pore volume, V_total_, was evaluated from the adsorption branch of the nitrogen adsorption-desorption isotherm for the maximum relative pressure p/p_0_ = 0.99. The mesopore-macropore-size distributions were evaluated using the Barret-Joyner-Halenda (BJH) method [32], using the Roberts algorithm [33] and the standard Broekhoff-deBoer isotherm with the Faas correction, using a Micromeritics software (3Flex, V6.02, Norcross, GA, USA).

Structural properties of catalytic foams were determined by X-ray diffraction (XRD), specifically the crystalline phases of oxides present and the size of the crystallites. The diffraction patterns were measured on a Rigaku SmartLab powder diffractometer using the Bragg-Brentano Θ–2Θ geometry of the goniometer (Rigaku, Wilmington, MA, USA). For the XRD measurements, the powder equivalents of the catalysts deposited on the foams (i.e., oxides’ powders without foam) were used. Each powder catalyst in a thin layer (0.5 mm) on a silicon wafer in the form of a tablet with a diameter of 24 mm was prepared for the XRD measurement. Diffraction recording was made in the range of 2Θ = 5–90° with a step of 0.01° using a D/teX Ultra 250 detector (Rigaku Technologies, Austin, TX, USA) in a continuous scanning mode and a CoKα lamp (λ1 = 1.78892 nm, λ2 = 1.79278 nm). Phase composition was determined using the reference diffractograms database ICDD (PDF-2). The size of the crystallites was calculated based on the Scherrer equation with a diffractometer resolution correction to LaB_6_ standard.

Temperature-programmed desorption of ammonia (NH_3_-TPD) was performed on an adsorption analyser AutoChem II (Micromeritics, Norcross, GA, USA) to investigate acid properties of the surface of the catalytic foams. The measurements were accomplished with two pieces of catalytic foams (mass of deposited catalyst ~60–66 mg) in the temperature range of 0–500 °C, with He as a carrier gas and NH_3_ as an adsorbing gas. Prior to the measurement, each catalytic foam was activated in He (50 mL/min) at 500 °C for 1 h, then cooled to 0 °C and saturated by 5% NH_3_/He for 30 min at 0 °C. Then, the catalytic foams were flushed with He (50 mL/min) to remove physically adsorbed NH_3_ up to the stable baseline (it took 110 min), with control using the mass spectrometer (MS-Hiden, Warrington, WA, United Kingdom). After that, the catalytic foams were heated in a He flow (50 mL/min) with a heating rate of 10 °C/min from 0 °C up to 500 °C (keeping at 500 °C for 5 min). A change in NH_3_ concentration was detected by a thermal conductivity detector (TCD, Norcross, GA, USA) and a mass spectrometer (MS). During the measurements, the following mass contributions were detected: 2-H_2_, 18-H_2_O, 16-NH_3_ and 44-CO_2_. TPD experiments were evaluated based on the calibration using a Micromeritics software (AutoChem II 2920MA, V6.01).

Temperature-programmed desorption of carbon dioxide (CO_2_-TPD) was performed on an adsorption analyser AutoChem II (Micromeritics, Norcross, GA, USA) as well to investigate basic properties of the surface of the catalytic foams. The measurements were accomplished with two pieces of catalytic foams (mass of deposited catalyst ~57–68 mg) in the temperature range of 0–500 °C, with He as a carrier gas and CO_2_ as an adsorbing gas. Prior to the measurement, each catalytic foam was activated in He (50 mL/min) at 500 °C for 1 h, then cooled to 0 °C and saturated by 50% CO_2_/He for 30 min at 0 °C. Then, the catalytic foams were flushed with He (50 mL/min) to remove physically adsorbed CO_2_ up to the stable baseline (it took 65 min), with control using a mass spectrometer (MS-Hiden). After that, the catalytic foams were heated in a He flow (50 mL/min) with a heating rate of 10 °C/min from 0 °C up to 500 °C (keeping at 500 °C for 5 min). A change in CO_2_ concentration was detected by a thermal conductivity detector (TCD) and a mass spectrometer (MS). During the measurements, the following mass contributions were detected: 2-H_2_, 18-H_2_O, 16-NH_3_ and 44-CO_2_. TPD experiments were evaluated based on the calibration using a Micromeritics software.

Temperature-programmed reduction by hydrogen (H_2_-TPR) was performed on an adsorption analyser AutoChem II (Micromeritics, Norcross, GA, USA) to investigate reducibility of the catalystic foams. The measurements were accomplished with two pieces of catalytic foams (mass of deposited catalyst ~58–65 mg) in the temperature range of 0–800 °C, with 10% H_2_/Ar mixture, Ar as a reference gas and H_2_ as a reducing gas. Prior to the measurement, each catalytic foam was dried at 250 °C for 30 min in Ar (30 mL/min), cooled to 28 °C and after setting the baseline the experiment started. The catalyst was heated in a 10% H_2_/Ar mixture flow (30 mL/min) with a heating rate of 10 °C/min from 28 °C up to 800 °C (keeping at 800 °C for 15 min). A change in the H_2_ concentration was detected by TCD. The TPR experiments were evaluated based on the calibration using a Micromeritics software.

For all measured temperature-programmed techniques (NH_3_-TPD, CO_2_-TPD and H_2_-TPR) the experimental error of measurements of temperature expressed as a relative standard deviation, determined on the basis of repeated tests of standards (Co_3_O_4_ and CoO), is 1.4%. The peaks area in NH_3_-TPD, CO_2_-TPD and H_2_-TPR spectra was calculated as the area under the curve in the region directly defined by the temperature without any deconvolution procedure.

To determine chlorine in the catalytic foams after dichloromethane oxidation the catalytic foam was crushed in a friction bowl, placed in an ampoule and weighed. After that 5 mL H_2_O was added and the ampoule was placed in an ultrasound bath for 15min. After centrifugation, 4 mL of liquid was taken and diluted to 1:1 with water in an IC sample box. The sample was analysed using an ion chromatograph Metrohm (type ECO IC, a carbonate system with a suppressor, Metrosep A5 250/4.4 mm column, Herisau, Switzerland). This analysis is assigned as DisMC (join dissolution and mass chromatography).

Scanning electron microscope (SEM) Tescan Vega with a Tungsten cathode and an energy-dispersive X-ray spectroscopy (EDS-EDAX) (Tescan, Brno, Czech Republic) were used for morphological investigations. Micrographs were obtained using the modes of secondary electrons (SE) and backscattered electrons (BSE) with an acceleration voltage of 30 KeV. Before imaging, materials were gold sputtered to ensure adequate electron conductivity.

### 2.3. TiO_2_, CeO_2_ and CuO-Supported VUKOPOR^®^A Ceramic Foams Testing in Methanol and Dichloromethane Oxidation

Testing of catalytic foams in methanol and dichloromethane oxidation was carried out in a fixed-bed flow tubular quartz reactor (inner diameter 8 mm) placed in a vertically situated ceramic tubular oven, operating under atmospheric pressure. The scheme of catalytic testing apparatus is shown in Figure 2. Temperature during the catalytic experiments was measured on the reactor wall before the catalyst bed and all catalytic tests were evaluated with respect to this measured temperature. The length of the catalyst bed was kept constant 10.8 cm. The gas phase analysis was performed using an online Gasmet DX-4000N FTIR gas analyser (Gasmet, Helsinki, Finland) which was calibrated to detect chlorinated hydrocarbons such as C_2_Cl_4_, C_2_HCl_3_, CH_3_Cl, CHCl_3_, CH_2_Cl_2_, COCl_2_, HCl, and oxyderivatives of hydrocarbons such as CO_2_, CO, HCOOH, HCOOCH_3_, CH_2_O, CH_3_OH, CH_4_ and C_2_H_4_. The measured IR spectra were analysed with a Calcmet software. The FTIR analyser is not able to detect noble gases and diatomic homonuclear compounds such as O_2_, N_2_, H_2_, and Cl_2_.

The initial methanol concentration was 500 ppm in all tests; i.e., ~0.715 g m^−3^ of methanol. Methanol was injected into an air flow by using automatic syringe pump and immediately evaporated to the air stream. The catalytic activity tests were carried out with fresh ceramic foams, keeping the constant length in the catalyst bed, i.e., 10.8 cm (the mass of deposited catalysts were as follows: TiO_2_-CeO_2_-CuO 0.1819 g, TiO_2_-CeO_2_ 0.1985 g, TiO_2_-CuO 0.1748 g), which were pretreated prior to each catalytic test. The pretreatment of the fresh catalytic foam consisted of heating up the catalytic foam in an air flow from laboratory temperature up to 500 °C and cooling down to 45 °C, when the catalytic test with methanol started. The catalytic test with the same catalytic foam was repeated 2–4 times. Air flow was 1.05 L min^−1^, which corresponded to the space velocity (GHSV) of 11, 600 h^−1^. During the catalytic experiment the reaction temperature was increased from 45 °C up to 400 °C with a heating rate 5 °C min^−1^.

Temperatures T_50_, T_90_, T_95_ and C_max_ corresponding to 50, 90, 95% and the maximum achieved methanol conversion were chosen as comparative measures of the catalyst activity. The CO_2_ and H_2_O yields were chosen as measures of catalyst selectivity.

The methanol conversion, C_MeOH_, was calculated according to Equation (1):(1)$$C_{\mathrm{MeOH}}=\frac{c_{\mathrm{MeOH},0}-c_{\mathrm{MeOH}}}{c_{\mathrm{MeOH},0}}\cdot 100$$
where C_MeOH_ is the methanol conversion (%), c_MeOH,0_ is the initial concentration of CH_3_OH (volume ppm) and c_MeOH_ is the measured output concentration of methanol (volume ppm).

The CO_2_ yield, Y_CO2,_ was calculated according to Equation (2):(2)$$Y_{\mathrm{CO}2}=\frac{c_{\mathrm{CO}2}}{c_{\mathrm{MeOH},0}}\cdot 100$$
where Y_CO2_ is the CO_2_ yield (%), c_CO2_ is the measured output concentration of CO_2_ (volume ppm) and c_MeOH,0_ is the initial concentration of methanol (volume ppm).

The H_2_O yield, Y_H2O,_ was calculated according to Equation (3):(3)$$Y_{H2O}=\frac{c_{H2O}}{2c_{\mathrm{MeOH},0}}\cdot 100$$
where Y_H20_ is the water yield (%), c_H2O_ is the measured output concentration of water (volume ppm) and c_MeOH,0_ is the initial concentration of methanol (volume ppm).

The initial dichloromethane concentration was 500 ppm in all tests, i.e., ~1.895 g m^−3^ of dichloromethane. The catalytic activity measurement was performed in the presence of 1.5 wt.% of water. Dichloromethane and water were simultaneously injected into an air flow by using automatic syringe pumps and were evaporated immediately. The catalytic activity tests were carried out with fresh catalytic foams, keeping the constant length of catalyst bed in 10.8 cm (the mass of the deposited catalyst was 0.1650 g for TiO_2_-CeO_2_-CuO, 0.2029 g for TiO_2_-CeO_2_, 0.1737 g for TiO_2_-CuO), which was pretreated prior to each catalytic test. The pretreatment of the fresh catalytic foam consisted of heating up the catalyst in an air flow from room temperature to 500 °C and cooling down to 100 °C when the catalytic test started. The catalytic test with the same catalytic foam was repeated 2–4 times. The air flow was 1.05 L min^−1^, which corresponded to the space velocity (GHSV) of 11, 600 h^−1^. During the catalytic experiment the reaction temperature was increased from 100 °C up to 500 °C with a heating rate 5 °C min^−1^.

Temperatures T_50_ and T_80_ and C_max_ corresponding to 50 and 80% dichloromethane conversion and the achieved maximum dichloromethane conversion were chosen as comparative measures of the catalyst activity. The maximum reached CO_2_ concentration (in ppm) and HCl yield were chosen as measures of selectivity.

The dichloromethane conversion, C_DCM_, was calculated according to Equation (4):(4)$$C_{\mathrm{DCM}}\left(\%\right)=\frac{c_{\mathrm{DCM},0}-c_{\mathrm{DCM}}}{c_{\mathrm{DCM},0}}\cdot 100$$
where C_DCM_ is dichloromethane conversion (%), c_DCM,0_ is the initial concentration of dichloromethane (volume ppm) and c_DCM_ in the measured output dichloromethane concentration (volume ppm).

The HCl yield, Y_HCl_, was calculated according to Equation (5):(5)$$Y_{\mathrm{HCl}}=\frac{c_{\mathrm{HCl}}}{2c_{\mathrm{DCM},0}}\cdot 100$$
where Y_HCl_ is the HCl yield (%), c_HCl_ is the measured output concentration of HCl (volume ppm) and c_DCM,0_ is the initial dichloromethane concentration (volume ppm).

The CO_2_ yield, Y_CO2_, was calculated according to Equation (6):(6)$$Y_{\mathrm{CO}2}=\frac{c_{\mathrm{CO}2}}{c_{\mathrm{DCM},0}}\cdot 100$$
where Y_CO2_ is the CO_2_ yield (%), c_CO2_ is the measured output concentration of CO_2_ (volume ppm) and c_DCM,0_ is the initial concentration of dichloromethane (volume ppm).

The stability test of the catalytic foams in methanol oxidation was performed with 6 pieces of the best catalytic foams, which created the catalyst bed of ~10.5 cm (corresponding to similar space velocity as in the light-off tests). Foams were first pre-treated in an air flow by heating-up them from room temperature up to 400 °C with heating rate of 5 °C min^−1^ and then cooled down to the starting temperature (i.e., to T_90_). After reaching T_90_ ~233 °C, the 20 h stability test was started with the methanol concentration of 500 ppm in an air flow of 1.05 L min^−1^. Based on the knowledge from the light-off tests, the main reaction products (CO_2_, H_2_O) and expected undesired by-products (CH_2_O, CO, HCOOH, CH_4_, HCOOCH_3_) were followed. The data was saved for each 60 s.

## 3. Results and Discussion

### 3.1. Physicochemical Characterization of TiO_2_, CeO_2_ and CuO-Supported VUKOPOR^®^A Ceramic Foams

**Chemical composition.** XRF results proving the chemical composition of the supported catalysts are summarized in Table 2. Results evidence that the TiO_2_-CeO_2_-CuO foam is composed of 86 wt.% TiO_2_, 11 wt.% of CeO_2_ and 3 wt.% of CuO, but the other two catalytic foams are composed of 93 wt.% of TiO_2_ and 7 wt.% of CeO_2_ (TiO_2_-CeO_2_ foam) and 97 wt.% of TiO_2_ and 3 wt.% of CuO (TiO_2_-CuO foam).

**Porous structure.** Concerning textural properties of the catalytic foams (Figure 3, Table 3), according to the shape of adsorption-desorption isotherms (Figure 3a) which are all typical IV type of the IUPAC isotherms, they are all mesoporous solids with narrow mesopores of 8–15 nm diameter (Figure 3b). It is evident that while the porous structures of the TiO_2_-CeO_2_-CuO and TiO_2_-CeO_2_ foams are very similar, showing similar specific surface areas and net pore volumes, the TiO_2_-CuO foam shows about 37% lower specific surface area, corresponding to a material with larger mesopores (Table 3).

**Crystal structure and crystallinity.** XRD patterns reveal the nanocrystalline character of all the catalytic foams (Figure 4, Table 4). Both TiO_2_-CeO_2_-CuO and TiO_2_-CeO_2_ foams are TiO_2_ anatase-CeO_2_ cerianite nanocrystalline structure, no crystalline phase of copper oxide/s was detected in the TiO_2_-CeO_2_-CuO foam. Copper is present in the TiO_2_-CeO_2_-CuO catalytic foam either as dispersed amorphous copper oxide/s, or as copper incorporated to the -O-Ti-O-Ce-O- structure in the Cu^2+^/Cu^+^ form. The TiO_2_-CuO foam was identified as a crystalline mixture of two TiO_2_ phases (anatase, rutile) and CuO tenorite, showing significantly larger anatase crystallites (~33 nm) than the TiO_2_-CeO_2_-CuO and TiO_2_-CeO_2_ foams (~12 nm and ~9 nm, Table 4).

**Redox properties.** In order to determine reducibility of the catalytic foams, the H_2_-TPR spectra were measured, and the overall reducibility of the catalytic foams was evaluated (Figure 5, Table 5). From the measured values of the consumed hydrogen, it is evident that the reducibility of the catalytic foams in the temperature range of 25–500 °C decreases in the following order: TiO_2_-CeO_2_-CuO foam > TiO_2_-CuO foam > TiO_2_-CeO_2_ foam (Table 5). In H_2_-TPR spectra of the catalytic foams for the temperature range of 25–500 °C, one main reduction peak with the maximum at 200 °C and a weak shoulder at 381°C was detected for the TiO_2_-CuO foam and one main reduction peak at 300 °C for the TiO_2_-CeO_2_-CuO foam. For the TiO_2_-CeO_2_ foam, there is a reduction peak starting at 400 °C with the maximum at 500–700 °C. The main reduction peak of the TiO_2_-CuO foam with a maximum at 200 °C and two weak side shoulders at 162 °C and 233 °C correspond to the reduction in Cu^2+^ → Cu^0^ within CuO, including three different forms of Cu^2+^ in CuO–Cu^2+^ in highly dispersed CuO species (at 162 °C), Cu^2+^ in bulk-like CuO (at 200 °C) and Cu^2+^ in a spinel form (probably Cu-O-Ti) (233 °C) [34]. The identified very weak reduction shoulder at 381 °C may be attributed to the reduction in Cu^+^ → Cu^0^ of a small amount of Cu_2_O present, which starts to reduce near to 300 °C [35] and its reduction may be finished at 600 °C (depending on the used temperature ramp during H_2_-TPR measurement [36]). These results are in agreement with the XRD results of the TiO_2_-CuO foam, where the CuO tenorite crystalline phase was identified. With respect to the very weak reduction peak of Cu_2_O, its very small amount in the TiO_2_-CuO foam can be assumed which is hard to detect by using XRD, taking into account the determined total copper oxide/s content ~3 wt.%. Concerning the reduction peaks of the identified TiO_2_ anatase and rutile in the TiO_2_-CuO foam, only weak reduction peaks can be seen at 500–700 °C corresponding to the reduction in TiO_2_ surface oxygen species. TiO_2_ reduction peaks typically appear at 500–750 °C and correspond to the reduction in TiO_2_ to nonstoichiometric TiO_2−x_. While peak/s at 500–620 °C correspond to the reduction in surface oxygen of TiO_2,_ the peak around 700 °C corresponds to the reduction in lattice oxygen of TiO_2_ [37,38]. Concerning the TiO_2_-CeO_2_ foam, only one broad reduction peak starting at 400 °C with maximum at 500–650 °C is observed. Typically, CeO_2_ shows low-temperature (300–570 °C) and high-temperature (600–850 °C) reduction peaks attributed to the reduction in surface oxygen species of CeO_2_ (300–570 °C), formation of nonstoichiometric cerium oxide/s (~580 °C) and bulk oxygen species of CeO_2_ (>600 °C), respectively, [38,39]. The appearance of one broad reduction peak starting at 400 °C with the maximum at 500–650 °C corresponds to the poorly nanocrystalline character of the TiO_2_-CeO_2_ foam [40], when strong interaction between Ce and Ti occurs. The presence of Ti ions can weaken the Ce–O bond which makes the reduction in Ce–O more facile [40,41]. Within the TiO_2_-CeO_2_-CuO foam, one main broad reduction peak at 300 °C with a shoulder at 370 °C arise. This peak may be attributed to the reduction in Cu^2+^ → Cu^0^ within CuO, including Cu^2+^ in the bulk-like CuO (at 300 °C) and Cu^2+^ in the spinel form (probably Cu-O-Ti) (at 370 °C). The shoulder at 370 °C can also correspond the reduction in Cu^+^ → Cu^0^ in Cu_2_O [35,36]. The shift of the copper oxide/s reduction peak to higher temperature (200 °C → 300 °C) will be a consequence of the strong interaction between Ce and Cu. Cu ions will weaken the Ce–O bond, making surface Ce–O easily reducible, therefore, the high-temperature reduction peak of ceria completely disappeared. This feature explains the high increase in the reducibility of the TiO_2_-CeO_2_-CuO foam in the temperature range of 25–500 °C. From XRD, the existence of nanocrystalline TiO_2_ anatase with the CeO_2_ cerianite nuclei was proved with no detected crystalline copper oxide phase/s. Thus, based on XRD and H_2_-TPR it can be assumed that the dominantly amorphous bulk phase CuO is dispersed close to the CeO_2_ nuclei in the surface layer of TiO_2_.

**Acidity.** The NH_3_-TPD spectra were measured in order to determine the acidity of the catalytic foams (Figure 6, Table 5). The NH_3_-TPD spectra and measured amounts of desorbed ammonia over all catalytic foams in the temperature range of 0–500 °C indicate that the total surface acidity of the catalytic foams decreases in the following order: TiO_2_-CeO_2_-CuO foam > TiO_2_-CeO_2_ foam > TiO_2_-CuO foam (Table 5). In general, the NH_3_-TPD spectra evidence dominant weak acid sites (at 0–200 °C), less medium strength acid sites (at 200–400 °C) and minimum of strong acid sites (at 400–500 °C), and this trend in the amounts follows the order of TiO_2_-CeO_2_-CuO foam > TiO_2_-CeO_2_ foam > TiO_2_-CuO foam. The positions of the main desorption peaks within 0–200 °C evidence that the TiO_2_-CuO foam shows the weakest acid sites with the maximum at 55 °C, and the TiO_2_-CeO_2_-CuO and TiO_2_-CeO_2_ foams show weak acid sites of more similar strength with the maxima at 58 °C and 63 °C, respectively. Thus, it is evident that the introduction of an amorphous bulk phase CuO to TiO_2_-CeO_2_, when the bulk phase CuO is in close interaction with the CeO_2_ nuclei within the TiO_2_ anatase structure, increases the overall catalyst acidity. This is an interesting feature, since in similar catalytic system, e.g., (Ce + Cu)/TiO_2_ from by Cao et al. [25], the addition of copper oxide decreased the acidity of the ternary catalyst compared to Ce/TiO_2_. Since the acidity is a beneficial catalyst property affecting its activity in dichloromethane oxidation via dissociative adsorption of Cl and the rupture of weak C–Cl bonds [25], it is worth to stress the positive effect of the used preparation method. Since copper provides copious Lewis acid sites and cerium provides both Lewis and Brønsted acid sites [19], then TiO_2_-CeO_2_-CuO will tend to show enhanced weak Lewis acidity, which is in an agreement with the acidity observations for a CuO-CeO_2_-TiO_2_ from by Chen et al. [19].

**Basicity**. CO_2_-TPD spectra were measured in order to determine the basicity of the catalytic foams (Figure 7, Table 5). The total amounts of desorbed CO_2_ for the temperature range of 0–500 °C show that the total surface basicity of the catalytic foams decreases in the following order: TiO_2_-CeO_2_-CuO foam > TiO_2_-CuO foam > TiO_2_-CeO_2_ foam (Table 5). The CO_2_-TPD spectra evidence dominant weak basic sites (at 0–200 °C), less medium strength basic sites (at 200–400 °C) and no strong basic sites (at 400–500 °C) (Figure 7). The TiO_2_-CuO foam possesses the highest abundance of weakest basic sites with the desorption peak maximum at 60 °C. TiO_2_-CeO_2_ shows two desorption peaks—a minor one with the maximum at 54 °C and a significant one with the maximum at 105 °C. The TiO_2_-CeO_2_-CuO foam also shows two desorption peaks—a small one with the maximum at 65 °C and a significant one with the maximum at 92 °C. It is evident that ~3 wt.% of CuO introduces to TiO_2_ the weakest basic sites, while ~7% wt.% of CeO_2_ introduces to TiO_2_ less weak basic sites. TiO_2_-CeO_2_-CuO, having both ~11% wt.% of CeO_2_ and ~3 wt.% of CuO in TiO_2_, shows an increased amount of the weakest (at 65 °C) as well as less weak (at 92 °C) basic sites compared to TiO_2_-CeO_2_. Introduction of CuO into TiO_2_-CeO_2_ creates more and weaker basic sites.

**Catalytic foams morphology.** SEM images show the macrostructure of the fresh ceramic foams with the deposited layer of the catalysts (Figure 8). It is evident from Figure 8a–f that the TiO_2_-CeO_2_-CuO catalytic foam is the smoothest, finest and the most compact with the minimum of catalytic layer cracks and defects (Figure 8a,b). The TiO_2_-CeO_2_ foam (Figure 8c,d) and the TiO_2_-CuO foam (Figure 8e,f) possess more defected and cracked layers of the catalysts. It is clear that the layer of the catalysts mostly covers the parent ceramic foam, and the catalytic layers are well-fixed on the parent ceramic foam.

### 3.2. Catalytic Results of TiO_2_, CeO_2_ and CuO-Supported VUKOPOR^®^A Ceramic Foams in the Oxidation of Methanol and Dichloromethane

**Methanol oxidation.** Light-off curves of methanol over all the investigated catalytic foams are shown in Figure 9a and the results of the catalytic activity and selectivity of the foams in methanol oxidation are summarized in Table 6, Figure 9b and Figure 10a–c. According to T_50_, T_90_ and T_95_, being the measures of catalyst activity, the catalytic activity of foams decreases as follows: TiO_2_-CuO foam~TiO_2_-CeO_2_-CuO foam > TiO_2_-CeO_2_ foam (Table 6). From the methanol conversion curves it is evident that 100% methanol conversion was reached only over the less active TiO_2_-CeO_2_ foam, while over the most active TiO_2_-CuO and TiO_2_-CeO_2_-CuO foams only 97% and 98% methanol conversions were reached, respectively (Figure 9a). Selectivity to CO_2_ of the foams decreases in the same order: TiO_2_-CuO foam > TiO_2_-CeO_2_-CuO foam > TiO_2_-CeO_2_ foam. The TiO_2_-CuO foam shows 100% CO_2_ selectivity, which is the best selectivity result over the studied foams. Concerning unwanted by-products during methanol oxidation on the TiO_2_-CuO foam, 20 ppm of CH_2_O (at 200 °C) and 2 ppm of CO (at 208 °C) were detected (Figure 10c). Both CeO_2_-containing catalytic foams (Figure 10a,b) show lower CO_2_ selectivity than the TiO_2_-CuO foam. Introduction of CeO_2_ (11 wt.%) to TiO_2_-CuO led to the decrease in the CO_2_ selectivity to 94%. The TiO_2_-CeO_2_ foam shows the poorest CO_2_ selectivity with high concentrations of unwanted by-products; CH_2_O (125 ppm) and CO (393 ppm) (Figure 9b). Based on the catalytic activity and selectivity results the TiO_2_-CuO foam can be supposed to be the best catalytic foam showing the best performance in methanol oxidation.

**Dichloromethane oxidation.** In Figure 11a the dichloromethane light-off curves over the investigated catalytic foams are shown and the results of the catalytic activity and selectivity of the foams in dichloromethane oxidation are summarized in Table 7, Figure 11b and Figure 12a–c. According to T_50_ and dichloromethane light-off curves, the activity of the foams is decreasing in the following order: TiO_2_-CeO_2_ foam > TiO_2_-CuO foam~TiO_2_-CeO_2_-CuO foam (Table 7). The reached 82–89% dichloromethane conversions and significantly higher T_50_ of all the investigated catalytic foams evidence their lower activity in dichloromethane oxidation than in methanol oxidation. The most active TiO_2_-CeO_2_ foam possesses the worst CO_2_ selectivity, detecting high concentrations of unwanted oxidation by-products such as 12 ppm of CH_2_O and 305 ppm of CO. The HCl yield is comparable for all the investigated catalytic foams in the range of 62–76% (Figure 11b). Positively, no chlorinated by-products were detected over the TiO_2_-CeO_2_ foam (Figure 12b). Contrary to that, both CuO-containing catalytic foams show significantly improved CO_2_ selectivity with detected traces of by-products (2–15 ppm of CO, 6–9 ppm of CHCl_3_) (Figure 12a,c). Overall, both CuO-containing catalytic foams show high CO_2_ selectivity, good HCl yield as well as the lowest detected traces of by-products. It is evident that copper oxide/s in the TiO_2_-based catalysts increases their CO_2_ selectivity and causes the formation of unwanted chlorinated by-products such as chloroform (CHCl_3_).

### 3.3. Effect of Physicochemical Properties of Catalytic Foams on Their Catalytic Performance in the Oxidation of Methanol

According to T_50_, T_90_ and T_95_, the catalytic activity of the foams decreases as follows: TiO_2_-CuO foam~TiO_2_-CeO_2_-CuO foam > TiO_2_-CeO_2_ foam (Table 6). Selectivity of the foams decreases in the order: TiO_2_-CuO foam > TiO_2_-CeO_2_-CuO foam > TiO_2_-CeO_2_ foam. The TiO_2_-CuO foam shows the best (100%) CO_2_ yield. Concerning the unwanted by-products during the methanol oxidation on the TiO_2_-CuO foam, 20 ppm of CH_2_O and 2 ppm of CO were detected (Figure 10c). Both CeO_2_-containing catalytic foams (Figure 10a,b) show lower selectivity than the TiO_2_-CuO foam. Introduction of CeO_2_ (11 wt.%) to TiO_2_-CuO led to a decrease in the CO_2_ yield, i.e., 94%. The TiO_2_-CeO_2_ foam shows the most poor CO_2_ yield, only 25%, with high concentrations of unwanted by-products; i.e., CH_2_O (125 ppm) and CO (393 ppm) (Figure 9b). Based on the catalytic activity and selectivity results, the TiO_2_-CuO foam possesses the best performance in methanol oxidation.

Tatibouët [42] reported interestingly that the by-product distribution along the catalytic methanol oxidation may be a good tool for the indication of catalyst surface acid-base properties. Formaldehyde (CH_2_O) is formed on both weak acid and basic sites when the H-abstraction is limited and strong adsorption of formaldehyde is prevented. If the acid sites are too strong, then dioxymethylene species are formed and they react with the neighbouring methoxy groups or adsorbed methanol and form methylal ((CH_3_O)_2_CH_2_). These reactions happen at low temperatures. When both acid and basic sites are stronger than those for methylal formation, the dioxymethylene species are oxidized into formate species which quickly react with methanol to form methyl formate (HCOOCH_3_) or are further oxidized to carbon oxides (CO, CO_2_). When strong acid sites and very weak basic sites are present, dimethyl ether ((CH_3_)_2_O) is formed. In summarization, the formation of dimethyl ether occurs with high acidic character of the catalyst, carbon oxides are formed at high basic character of the catalyst and mild oxidation products are formed at bifunctional acid-base character of the catalyst [42]. In our case, formaldehyde (CH_2_O) and carbon oxides (CO, CO_2_) were identified as the only reaction products. Formic acid (HCOOH), methyl formate (HCOOCH_3_) or methane (CH_4_), which was possible to detect by the calibrated FTIR, were not detected at all during the methanol oxidation over any of the catalytic foams. In oxygen deficiency, hydrogen (H_2_) may be produced, but in our case in the oxygen-rich atmosphere water (H_2_O) is the reaction product. Methylal ((CH_3_O)_2_CH_2_) was not possible to be analysed by FTIR because of missing calibration. The measured selectivity results correspond very well to the fact that all the tested catalytic foams possess weak basic and weak-to-medium strength acidic sites which revealed acidity and basicity results summarized in Table 5, Figure 6 and Figure 7. However, there exist some differences, which result in different reaction products distribution and CO_2_ selectivity of the individual catalytic foams (Figure 9b and Figure 10a–c, Table 6). The introduction of amorphous bulk phase CuO to TiO_2_-CeO_2_, when amorphous bulk phase CuO is in close contact with CeO_2_ nuclei within the TiO_2_ anatase structure, increased significantly (2x) the weak Lewis acidity as well as the weak basicity of the TiO_2_-CeO_2_-CuO foam (0.36 mmol_NH3_ g^−1^, 0.23 mmol_CO2_ g^−1^) compared to the TiO_2_-CuO foam (0.20 mmol_NH3_ g^−1^, 0.20 mmol_CO2_ g^−1^). Moreover, the maximum of the CO_2_-desoption peak of the TiO_2_-CeO_2_-CuO foam was shifted to higher temperature of 92 °C compared to the TiO_2_-CuO foam (60 °C), indicating not only higher abundance but also the presence of stronger weak basic sites. Both increased weak Lewis acidity and stronger basicity of the TiO_2_-CeO_2_-CuO catalytic foam resulted in stronger adsorption of formate species which are further oxidized to the mixture of CO and CO_2_. In a consequence of increased formate adsorption due to higher amount of weak Lewis acid sites and increased stronger basicity, lower selectivity to CO_2_ was reached on the TiO_2_-CeO_2_-CuO catalytic foam. It shows that the TiO_2_ anatase-CuO tenorite crystalline structure of the TiO_2_-CuO foam, which shows the lowest Lewis acidity and the weakest basic sites is advantageous for the selective methanol oxidation to CO_2_. This is in agreement with the reported aspect that transformation of adsorbed methoxy group depends on the strength of the acid site on which methoxy group is adsorbed and on the nature of the active centres in close proximity. The ability to break a C-H bond, being the reaction rate determining step, depends on the basic or nucleophilic character of oxygen species close to the methoxy group [42]. Thus, bifunctional acid-base character of the catalyst plays the key role in catalyst performance (activity as well as selectivity) in methanol oxidation.

The second important property is the catalyst reducibility, since when the catalyst shows its high reducibility at low temperature, it may effectively give electrons and oxidize on its active surface sites the adsorbate at low temperatures. Despite the TiO_2_-CeO_2_-CuO foam shows higher reducibility than the TiO_2_-CuO foam, the maximum of its broad reduction peak is shifted to higher temperature of 300 °C contrary to the TiO_2_-CuO foam, which maximum of reduction peak is located at lower temperature of 200 °C (Figure 5). This shift of the copper oxide/s (mainly CuO) reduction peak to higher temperature (200 °C → 300 °C) is a consequence of the strong interaction between Ce and Cu, when surface Ce–O are more easily reducible in the TiO_2_-CeO_2_-CuO foam. It is evident that good low-temperature reducibility of copper oxide/s (mainly CuO) in the TiO_2_-CuO foam promotes the selectivity to CO_2_. The importance of catalyst reducibility evidences the case of the TiO_2_-CeO_2_ foam, which shows medium weak acidity (0.27 mmol_NH3_ g^−1^, Table 5), but a low amount of weak basic sites (0.18 mmol_CO2_ g^−1^, CO_2_-desorption peak maximum at 103 °C, similar basic sites as in the TiO_2_-CeO_2_-CuO foam). Thus, the rate of transformation of the adsorbed methoxy species to the formate species, which are quite strongly adsorbed is very slow due to minimum contribution of adjacent basic or nucleophilic character of the oxygen species. The poor reducibility (0.54 mmol_H2_ g^−1^) of the TiO_2_-CeO_2_ foam within the temperature range of 25–400 °C does not contribute to the improvement of the CO_2_ yield as well, being maximally 25% accompanied with the production of ~380–400 ppm of CO at 100% methanol conversion. The desorption of the reaction products is more favoured by weak than by strong acid sites [43]. But, in the case of the tested catalytic foams the distribution of the types of acid sites is very similar (Figure 6) contrary to basic sites (Figure 7) and redox properties (Figure 5). Thus, it is evident that the activity and selectivity of the catalytic foams in methanol oxidation are strongly influenced besides the catalyst acido-basic nature also by the reducibility of active species of the catalyst (Figure 13). This is in agreement with the reported fact that the catalyst activity is given by basic or nucleophilic character of oxygen species close to adsorbed methoxy group, being able to break a C–H bond [42]. It can be concluded that due to low weak Lewis acidity, high weak basicity and easily reducible CuO species the TiO_2_-CuO foam shows the best catalytic performance in methanol oxidation.

### 3.4. Effect of Physicochemical Properties of Catalytic Foams on Their Catalytic Performance in the Oxidation of Dichloromethane

Cao et al. [18,25] and El Assal et al. [26] reported interesting results concerning the dichloromethane oxidation over TiO_2_, CeO_2_ and CuO-based catalysts. As reported by Cao et al. [25], over a two-step Ce/TiO_2_-Cu/CeO_2_ catalytic system in dichloromethane degradation mechanism takes place basically in three steps: (1) dichloromethane adsorption and rupture of the C–Cl bonds, (2) deep oxidation of C–H from by-products and (3) total oxidation of CO to CO_2_. The advantage of their two step catalytic system is that the 1st and 2nd steps are physically separated from the 3rd one, avoiding the decrease in acid sites taking place on the (Ce+Cu)/TiO_2_ catalyst and the chlorine poisoning of TiO_2_ due to the strong adsorption of Cl on CuO [25]. They reported that in Ce/TiO_2_ chlorine is adsorbed on the surface oxygen vacancies of CeO_2_, preventing TiO_2_ (Ti^4+^ sites) deactivation by adsorbed chlorine, accompanied by CeO_2_ capability to enhance further deep oxidation of C–H from by-products and partial oxidation of CO to CO_2_ [18]. In fact, they are saying that the Ce/TiO_2_-Cu/CeO_2_ catalytic system in a series should be more efficient than the mixed ternary TiO_2_-CeO_2_-CuO catalyst. However, despite of their conclusions about the effective role of individual catalysts/oxides in the mechanism of dichloromethane degradation, they did not report on the existing particular chlorinated products and by-products selectivity besides CO + CO_2._ There is a lack of important information about selectivity to chlorinated products/by-products despite of the fact that this issue is crucial for developing selective catalysts for industrial chlorinated VOCs catalytic combustion since these harmful intermediates are part of the flue gas. Concerning chlorinated products/by-products selectivity over CuO-containing catalysts in dichloromethane oxidation under wet conditions Matějová et al. [27] observed some chloroform (CHCl_3_) formation over commercial Cu-Mn/Al_2_O_3_. El Assal et al. [26] observed the formation of chloroform (CHCl_3_) and chloromethane (CH_3_Cl) besides HCl and Cl_2_ over CuO-TiO_2_ and CuO-CeO_2_ catalysts. In the work of Matějová et al. [27], chloroform (CHCl_3_) was formed/observed at elevated temperature (>300 °C) for Cu-Mn/Al_2_O_3_ and the CuMnO_x_ phase was responsible for chloroform formation similarly as in [44], but for CuO-TiO_2_ and CuO-CeO_2_ in El Assal et al. work [26] the cause of formation of chlorinated by-products was not discussed. It may be assumed that both chloromethane and chloroform over CuO-TiO_2_ and CuO-CeO_2_ in wet conditions may be formed as follows: The reaction of CH_3_O-species formed/adsorbed on acid sites with formed HCl or Cl_2_ takes place, forming CH_3_Cl. Chlorine adsorbed on surface CeO_2_ oxygen vacancies (in CuO-CeO_2_) or TiO_2_ acid sites (in CuO-TiO_2_) reacts with adsorbed CH_2_OCl-species, forming CHCl_3_. Moreover, higher HCl yield over CuO-TiO_2_ than over CuO-CeO_2_ in El Assal et al. work [26] evidences the importance of a higher amount of acid sites for increased selectivity to HCl. These results are indirectly complemented with conclusions of Cao et al. [25] who announced that Cu/CeO_2_ must be exposed to reduce the chlorine amount to reach improved CO_2_ selectivity.

The best CO_2_ and HCl yields (91% and 76%, respectively) reached at ~80% conversion of dichloromethane over the ternary TiO_2_-CeO_2_-CuO@VUKOPOR^®^A ceramic foam in this study reveal that by using a proper sol-gel method and at set chemical composition (Table 1) it is possible to prepare an efficient catalyst with suitable oxide phases’ overlap, similarly as in the model two-step Ce/TiO_2_-Cu/CeO_2_ catalytic system. Characterization results revealed that in the surface layer of -O-Ti-O- anatase crystal structure Ce^3+^/Ce^4+^ is incorporated which is in a close contact with the dispersed bulk phase CuO. Thus, within the TiO_2_-CeO_2_-CuO@VUKOPOR^®^A foam there are highly-reducible -O-Cu-Ce-O- active surface sites (1.07 mmol_H2_ g^−1^) as well as well-acidic -Ti-O-Ce-O- active sites (0.36 mmol_NH3_ g^−1^) which ensure both the good CO_2_ and HCl yields (Figure 14), also with respect to minimum formation of undesired by-products (only 15 ppm of CO and 6 ppm of CHCl_3_ was formed). While acid sites are important in adsorption of dichloromethane and surface lattice oxygen species in C–Cl and C–H bonds dissociation, the surface adsorbed oxygen species are responsible for the oxidation of CO to CO_2_. Since traces of CHCl_3_ were formed exclusively over both CuO-containing catalytic foams (TiO_2_-CeO_2_-CuO@VUKOPOR^®^A and TiO_2_-CuO@VUKOPOR^®^A, Table 7, Figure 12a,c) at temperatures above 400 °C, its formation may be attributed to the presence of CuO bulk phase. For TiO_2_-CeO_2_-CuO@VUKOPOR^®^A chlorine is firstly adsorbed on the oxygen vacancies of the Ce/TiO_2_(-Ti-υO-Ce-O-) active sites, but when these centres are completely occupied then it is adsorbed on CuO. For TiO_2_-CuO@VUKOPOR^®^A the adsorption of chlorine is more pronounced, reflected by the worst selectivity to CO_2_ and HCl and higher production of CHCl_3_. This agrees with chlorine amount identified in the catalytic foams after the dichloromethane oxidation tests (see next Section 3.5).

### 3.5. Chlorine in the Catalytic Foams after Dichloromethane Oxidation Testing

All catalytic foams after dichloromethane light-off tests were analysed by two techniques to determine the present amount of chlorine deposited: (1) by dissolution of chlorine in the catalytic foams and those solutions were subsequently analysed by using mass chromatography (DisMC) and (2) by surface analysis by using EDX-SEM.

From the DisMC analysis, which is the technique determining also bulk chlorine, the highest chlorine amount was identified in the TiO_2_-CeO_2_ foam (20.4 µg_Cl_ g^−1^_cat_) followed by the TiO_2_-CuO foam (9.5 µg_Cl_ g^−1^_cat_) (Table 8). From surface analysis by EDX-SEM the highest amount of chlorine was detected on the surface of the TiO_2_-CuO foam (1.12 wt.%) followed by the TiO_2_-CeO_2_ foam (0.72 wt.%) (Table 8, Figure 15). From the EDX-SEM images it is visible that Cl is present preferentially on the surface places with TiO_2_ and/or CeO_2_ in the TiO_2_-CeO_2_-CuO foam (Figure 15a). Concerning the TiO_2_-CeO_2_ foam it is not possible to distinguish (Figure 15b). In the case of the TiO_2_-CuO foam Cl is more present on places with TiO_2_ (Figure 15c).

The results of both analyses indicate that some chlorine stays after dichloromethane oxidation in the catalytic foams and this effect is the least pronounced for the TiO_2_-CeO_2_-CuO foam. The higher bulk chlorine content in the TiO_2_-CeO_2_ foam may correspond with its higher surface area (58 m^2^ g^−1^) and pore volume than those of the TiO_2_-CuO foam (37 m^2^ g^−1^). The lowest detected chlorine amount on the TiO_2_-CeO_2_-CuO foam proves the advantageous catalyst structure achieved by the used sol-gel technique.

### 3.6. Stability Study of TiO_2_-CuO@VUKOPOR^®^A Foam in Methanol Oxidation

TiO_2_-CuO foam, determined as the most active and selective catalytic foam for methanol oxidation, was selected for the stability test. The 20 h stability test of the TiO_2_-CuO foam in methanol oxidation at T_90_ ~233 °C (Figure 16) confirmed its good catalytic activity as well as stability, keeping slightly higher (than in light-off tests) and constant 90–95% methanol conversion for the whole testing time period (Figure 16a). The selectivity of the TiO_2_-CuO foam to main product during the stability test was also satisfactory, reaching 80–93% CO_2_ yield, which was slightly lower than during the light-off test when 95–97% CO_2_ yield (Figure 9b) was reached. Concerning the unwanted by-products at T_90_, CH_2_O and CO concentrations changed in the range of 5–8 ppm and 3–4 ppm during the stability test (Figure 16b), respectively. This is comparable as during the light-off test when 5 ppm of CH_2_O and 2 ppm of CO were detected (Figure 10c). Results from the 20 h stability testing are in good agreement with the light-off testing. It can be concluded that incomplete methanol conversion reached in the light-off tests (only 98% for TiO_2_-CeO_2_-CuO@VUKOPOR^®^A foam, 97% for TiO_2_-CuO@VUKOPOR^®^A foam) is a negative result of dynamic thermal process and it is within the measurement error of ±5%. Higher methanol conversion (90–95%) reached in the stability test than in the light-off test (90%) with the TiO_2_-CuO@VUKOPOR^®^A foam evidence that methanol can be completely oxidized.

The porous structure properties and carbon content in TiO_2_-CuO@VUKOPOR^®^A foam after 20 h stability test were examined. The specific surface area of 29 m^2^ g^−1^ and net pore volume of 92 mm^3^_liq_ g^−1^ were slightly decreased compared to fresh catalytic foam (37 m^2^ g^−1^, 119 mm^3^_liq_ g^−1^). However, no carbon content was detected in used TiO_2_-CuO@VUKOPOR^®^A foam. Thus, the slight decrease in specific surface area and net pore volume of used catalytic foam was not a consequence of formed carbon deposits and this feature was not reflected negatively to catalytic performance and stability of designed TiO_2_-CuO@VUKOPOR^®^A foam.

## 4. Conclusions

TiO_2_-CeO_2_-CuO, TiO_2_-CeO_2_ and TiO_2_-CuO@VUKOPOR^®^A foams were synthesized by using a simple reverse micelles-assisted sol-gel method and dip-coating methods, accompanied by calcination. This ceramic foam supported form of catalysts was investigated in the oxidation of two VOCs, methanol and dichloromethane, which are two most used solvents in pharmaceutical industry and analytical practise, thus, being mixed in gaseous industrial emissions. The used sol-gel method showed a good potential to be used for the preparation of effective structured form of appropriate catalyst structure for catalytic oxidation of two VOCs. The TiO_2_-CuO@VUKOPOR^®^A foam showed the best catalytic performance in methanol oxidation and the TiO_2_-CeO_2_-CuO@VUKOPOR^®^A foam was the best in dichloromethane oxidation. In methanol oxidation, low weak Lewis acidity, high weak basicity and easily reducible CuO species within the TiO_2_-CuO@VUKOPOR^®^A foam are responsible for the high activity and CO_2_ selectivity of the catalytic foam. The TiO_2_-CuO@VUKOPOR^®^A foam proved a good stability within 20 h methanol oxidation test. In dichloromethane oxidation, a high reducibility of the CuO-supported@VUKOPOR^®^A foams are responsible for their high CO_2_ selectivity. The appropriate crystal structure of TiO_2_-CeO_2_-CuO@VUKOPOR^®^A foam with highly-reducible -O-Cu-Ce-O- active surface sites led to the best CO_2_ yield in dichloromethane oxidation. The effect of weak basicity (surface lattice oxygen species) of the investigated catalytic foams on their catalytic activity or selectivity is not evident in the dichloromethane oxidation. Weak Lewis acidity, being important for the initial adsorption of dichloromethane, affects the HCl yield. Despite of the acidity of the catalytic foams in this study is not so high and does not differ significantly, the highest HCl yield was observed for the TiO_2_-CeO_2_-CuO@VUKOPOR^®^A foam showing the highest abundance of weak Lewis acid sites. This catalytic foam also showed the lowest amount of chlorine deposits after dichloromethane oxidation tests.

Findings presented within this study indicate a good potential of used sol-gel route for preparation of deposited thin film catalysts on the ceramic types of supports suitable for catalytic oxidation of VOCs in industrial scale.

Future work should be focused on the study of the effect of chemical composition (Ti:Ce:Cu molar ratio) of the deposited catalyst on its catalytic performance in individual VOCs oxidation. From more general point of view, the catalyst design and VOCs catalytic oxidation should follow the development direction that unwanted oxidation by-products are minimized and simultaneously NOx is not produced. These are the reasons why catalysts needs further research and each VOC emissions needs to be looked at.

## Figures and Tables

**Figure 1 nanomaterials-13-01148-f001:**
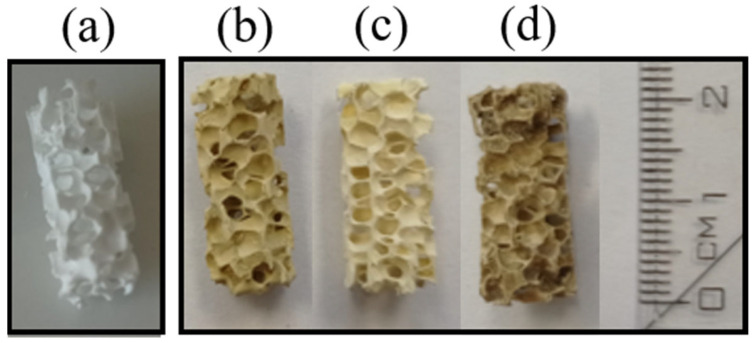
Images of (**a**) parent VUKOPOR^®^A ceramic foam, and the prepared (**b**) TiO_2_-CeO_2_-CuO, (**c**) TiO_2_-CeO_2_ and (**d**) TiO_2_-CuO-supported VUKOPOR^®^A ceramic foams.

**Figure 2 nanomaterials-13-01148-f002:**
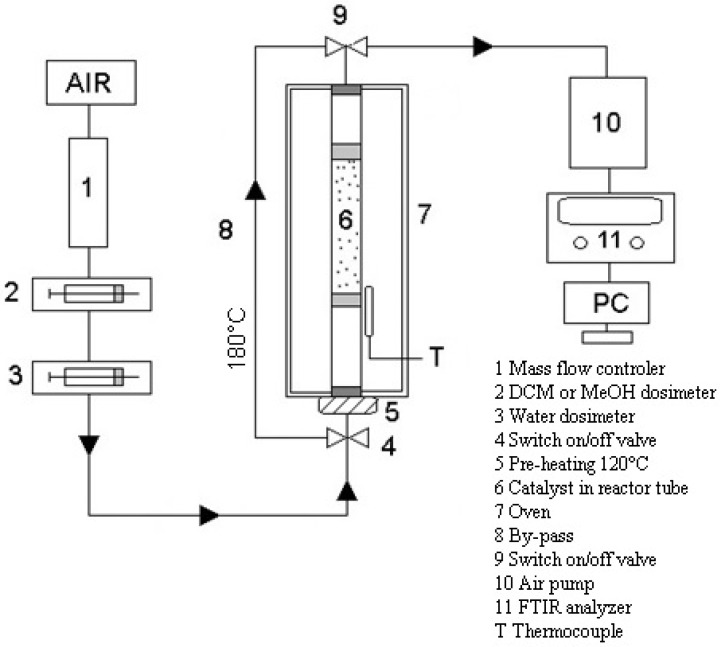
Scheme of the catalyst testing apparatus for VOCs oxidation.

**Figure 3 nanomaterials-13-01148-f003:**
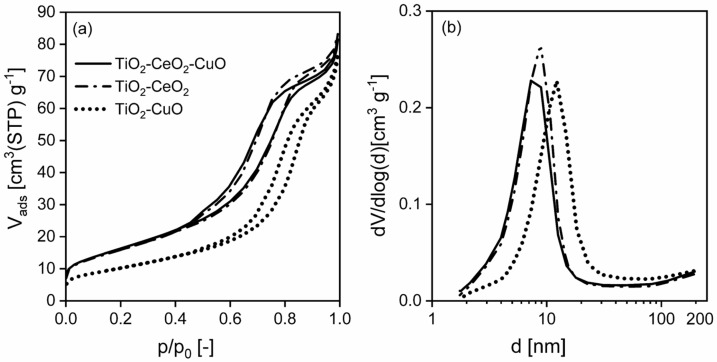
(**a**) Measured nitrogen adsorption-desorption isotherms at 77 K and (**b**) evaluated mesopore-macropore-size distributions of the investigated TiO_2_, CeO_2_ and CuO-supported VUKOPOR^®^A ceramic foams.

**Figure 4 nanomaterials-13-01148-f004:**
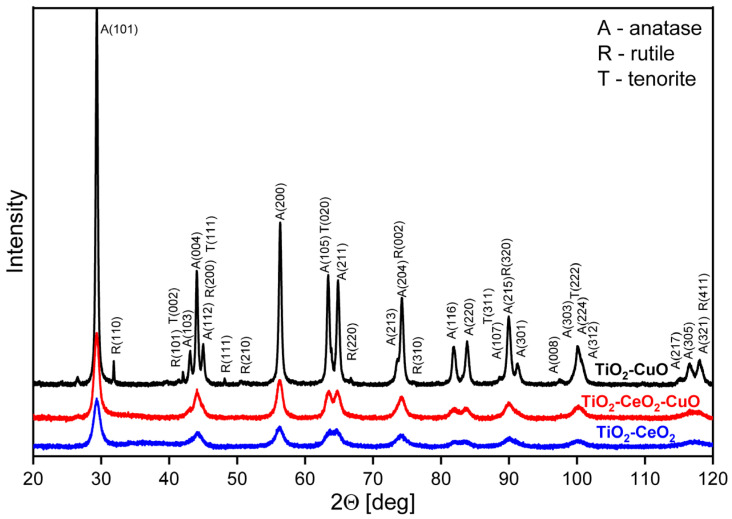
X-ray diffraction patterns of active phases of the investigated TiO_2_, CeO_2_ and CuO-supported VUKOPOR^®^A ceramic foams.

**Figure 5 nanomaterials-13-01148-f005:**
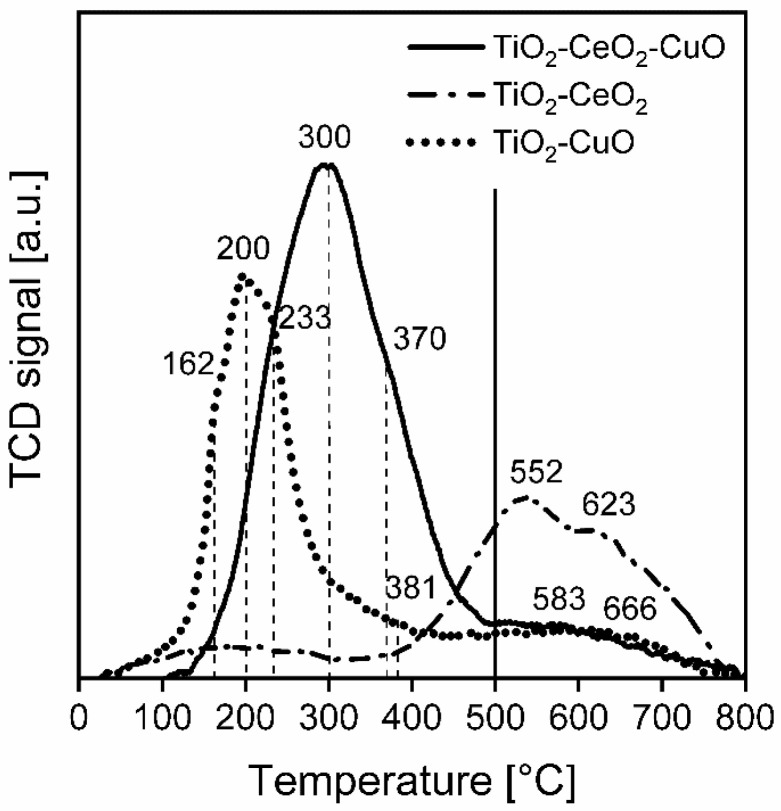
H_2_-TPR spectra of the investigated TiO_2_, CeO_2_ and CuO-supported VUKOPOR^®^A ceramic foams. Dashed vertical lines indicate the maxima of the individual identified reduction peaks. Solid vertical line indicates the calcination temperature used for catalytic foams preparation.

**Figure 6 nanomaterials-13-01148-f006:**
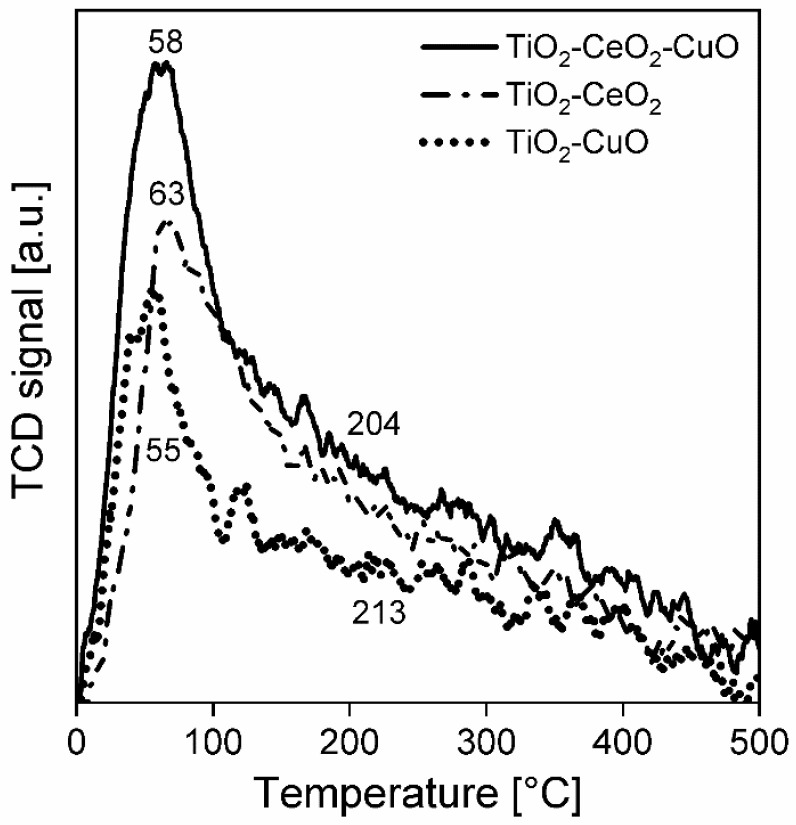
NH_3_-TPD spectra of the investigated TiO_2_, CeO_2_ and CuO-supported VUKOPOR^®^A ceramic foams.

**Figure 7 nanomaterials-13-01148-f007:**
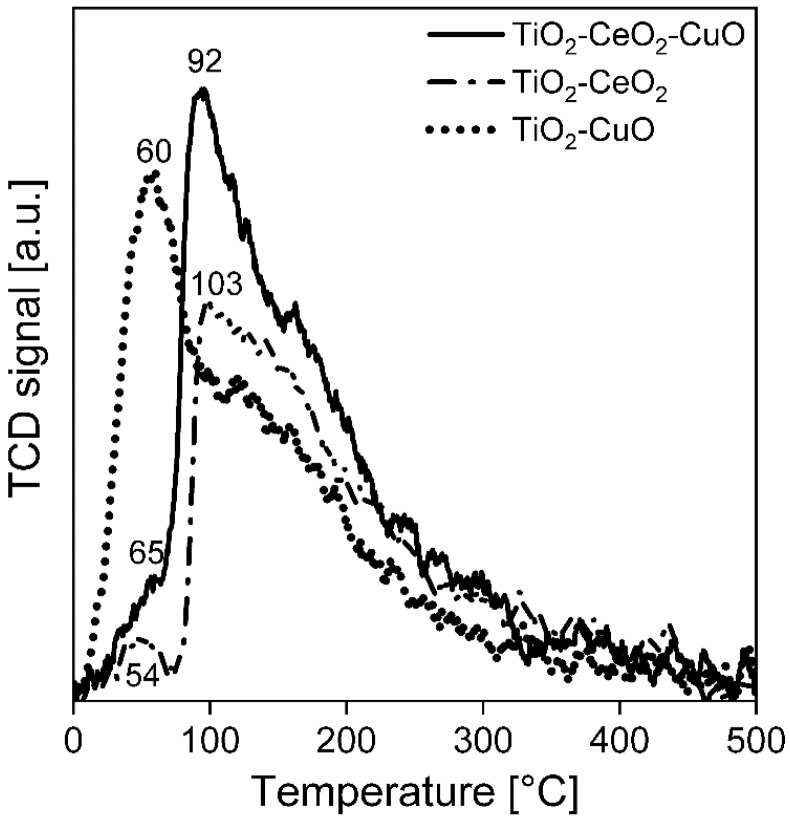
CO_2_-TPD spectra of the investigated TiO_2_, CeO_2_ and CuO-supported VUKOPOR^®^A ceramic foams.

**Figure 8 nanomaterials-13-01148-f008:**
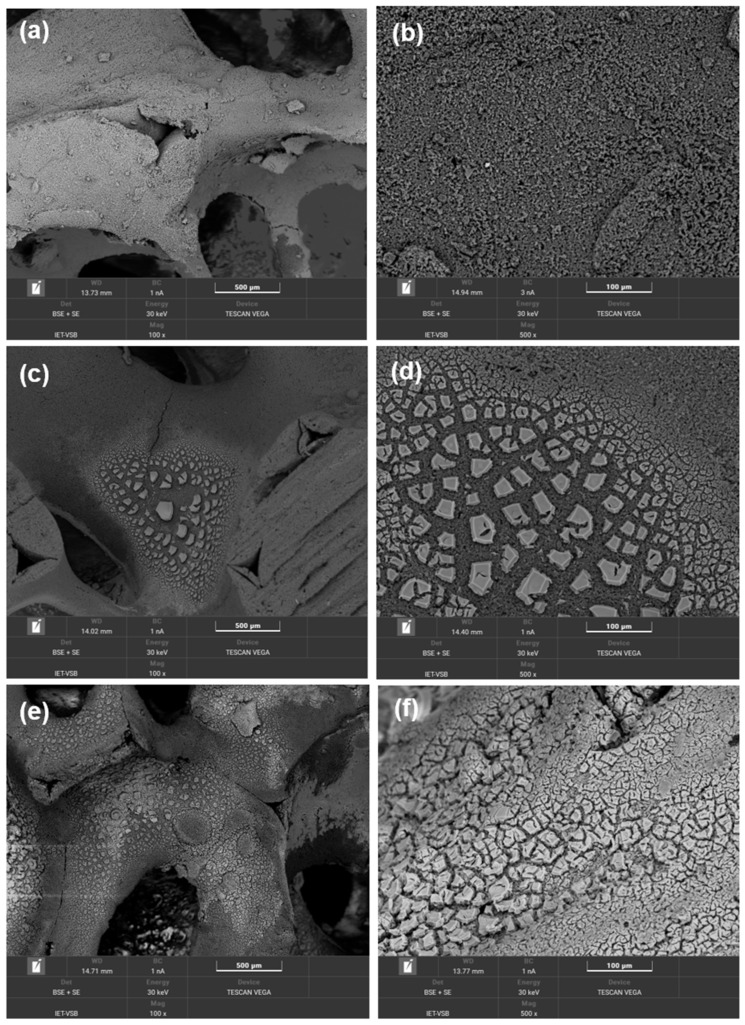
SEM images of the fresh catalytic foams: (**a**,**b**) TiO_2_-CeO_2_-CuO@VUKOPOR^®^A foam, (**c**,**d**) TiO_2_-CeO_2_@VUKOPOR^®^A foam and (**e**,**f**) TiO_2_-CuO@VUKOPOR^®^A foam.

**Figure 9 nanomaterials-13-01148-f009:**
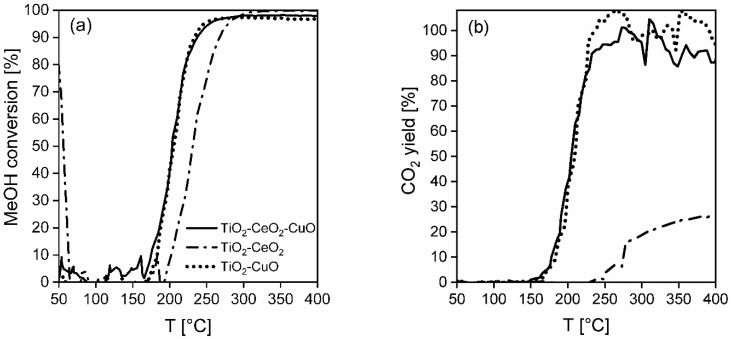
(**a**) Light-off curves of methanol and (**b**) CO_2_ yields in methanol oxidation over the investigated TiO_2_, CeO_2_ and CuO-supported VUKOPOR^®^A ceramic foams.

**Figure 10 nanomaterials-13-01148-f010:**
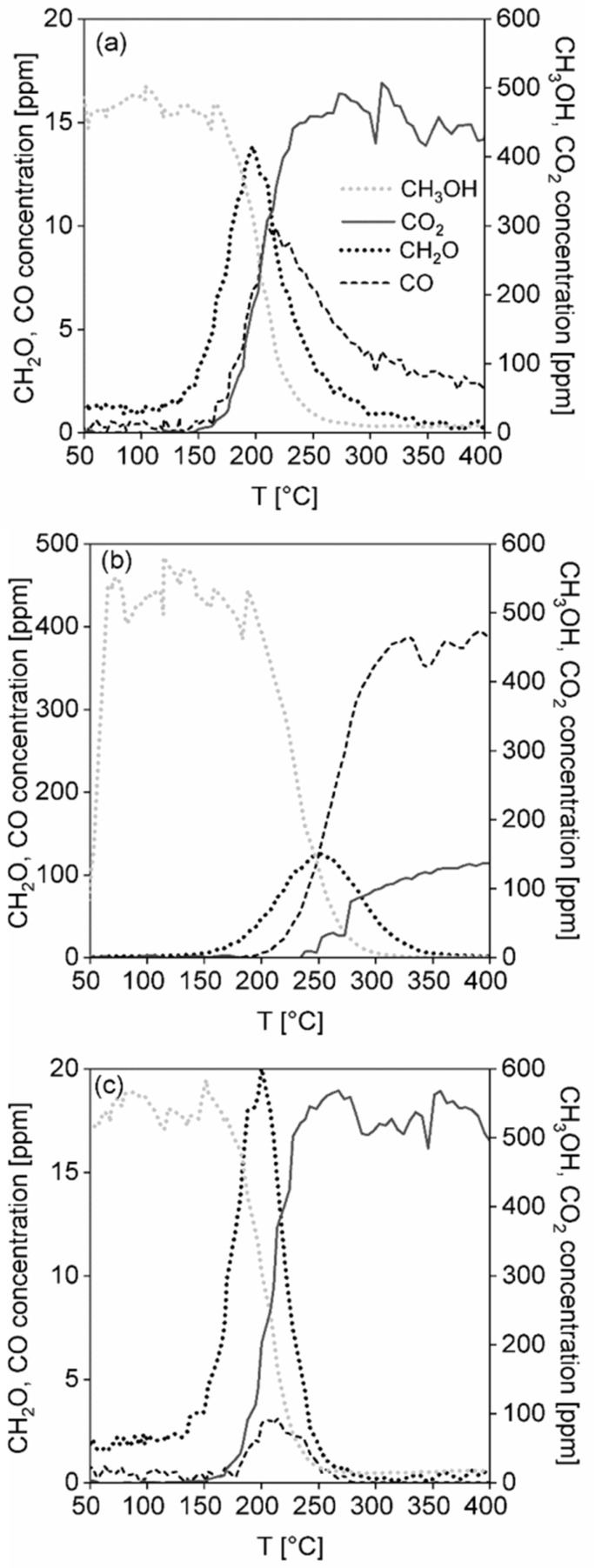
Concentration profiles of methanol and all detected main products and by-products in methanol oxidation over (**a**) TiO_2_-CeO_2_-CuO@VUKOPOR^®^A foam, (**b**) TiO_2_-CeO_2_@VUKOPOR^®^A foam and (**c**) TiO_2_-CuO@VUKOPOR^®^A foam.

**Figure 11 nanomaterials-13-01148-f011:**
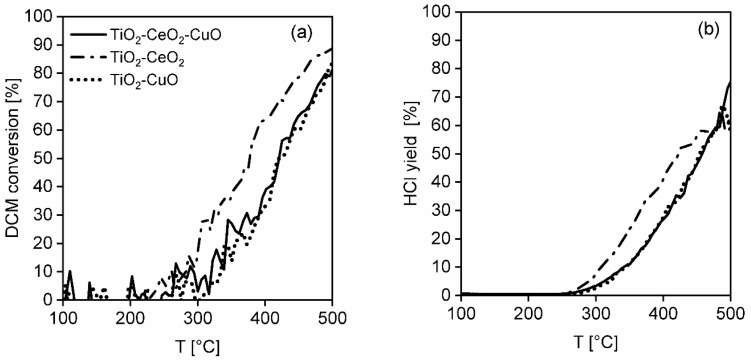
(**a**) Light-off curves of dichloromethane and (**b**) HCl yields in dichloromethane oxidation over the investigated TiO_2_, CeO_2_ and CuO-supported VUKOPOR^®^A ceramic foams.

**Figure 12 nanomaterials-13-01148-f012:**
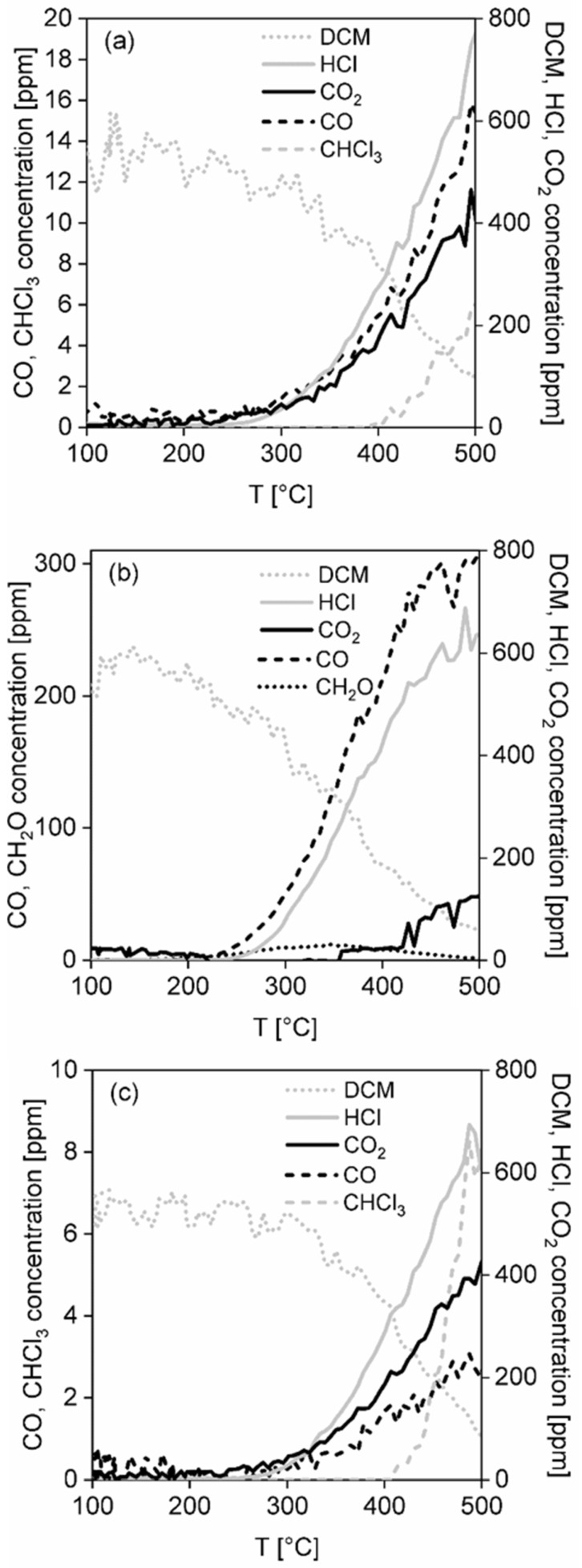
Concentration profiles of dichloromethane and all detected main products and by-products in dichloromethane oxidation over (**a**) TiO_2_-CeO_2_-CuO@VUKOPOR^®^A foam, (**b**) TiO_2_-CeO_2_@VUKOPOR^®^A foam and (**c**) TiO_2_-CuO@VUKOPOR^®^A foam.

**Figure 13 nanomaterials-13-01148-f013:**
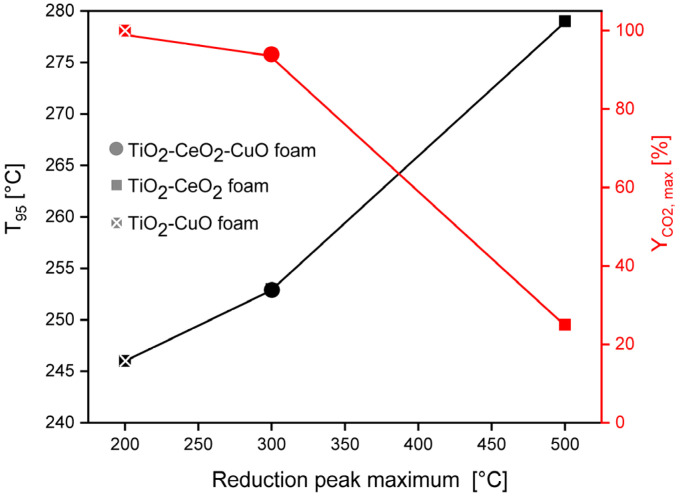
Correlation between the maximum of H_2_-TPR peak of active species, CO_2_ yield, and activity defined by T_95_ of the investigated TiO_2_, CeO_2_ and CuO-supported VUKOPOR^®^A ceramic foams in methanol oxidation.

**Figure 14 nanomaterials-13-01148-f014:**
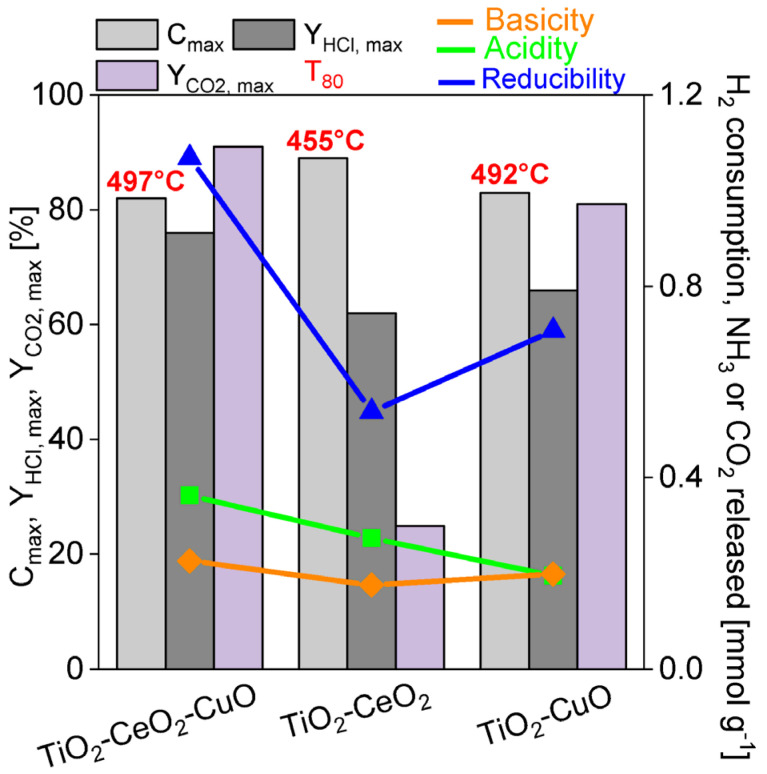
Correlation between acidity, basicity, reducibly and CO_2_ and HCl yields of the investigated TiO_2_, CeO_2_ and CuO-supported VUKOPOR^®^A ceramic foams in dichloromethane oxidation.

**Figure 15 nanomaterials-13-01148-f015:**
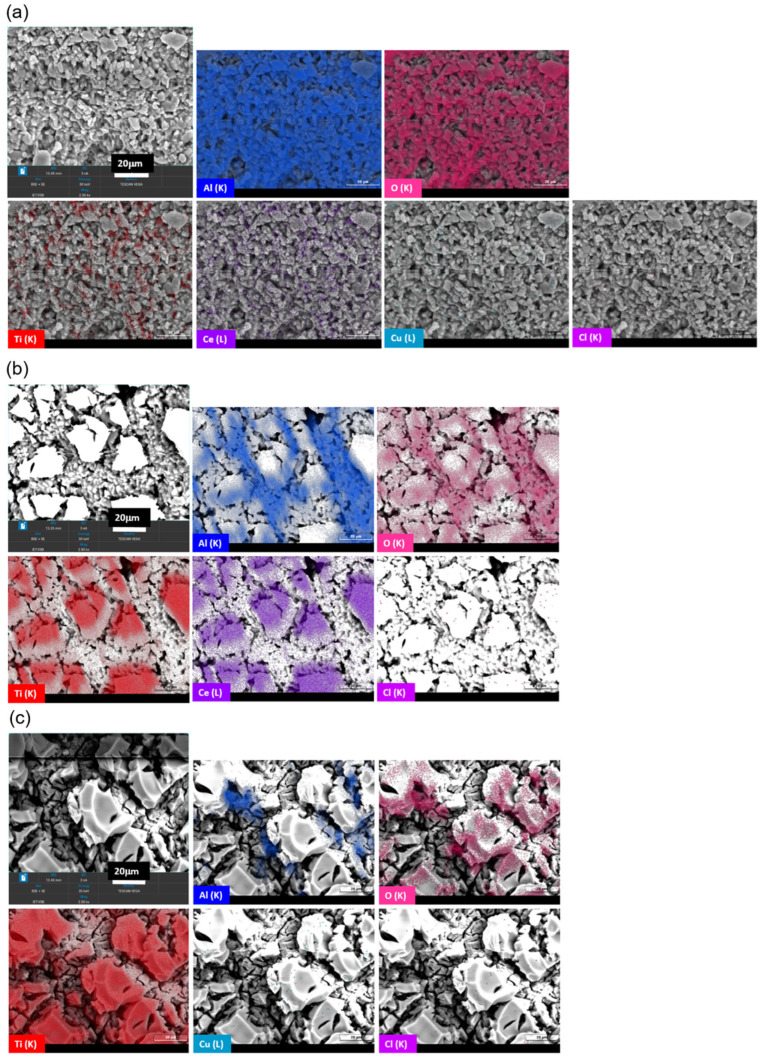
SEM images including elemental composition of the catalytic foams after dichloromethane oxidation (**a**) TiO_2_-CeO_2_-CuO@VUKOPOR^®^A foam, (**b**) TiO_2_-CeO_2_@VUKOPOR^®^A foam and (**c**) TiO_2_-CuO@VUKOPOR^®^A foam.

**Figure 16 nanomaterials-13-01148-f016:**
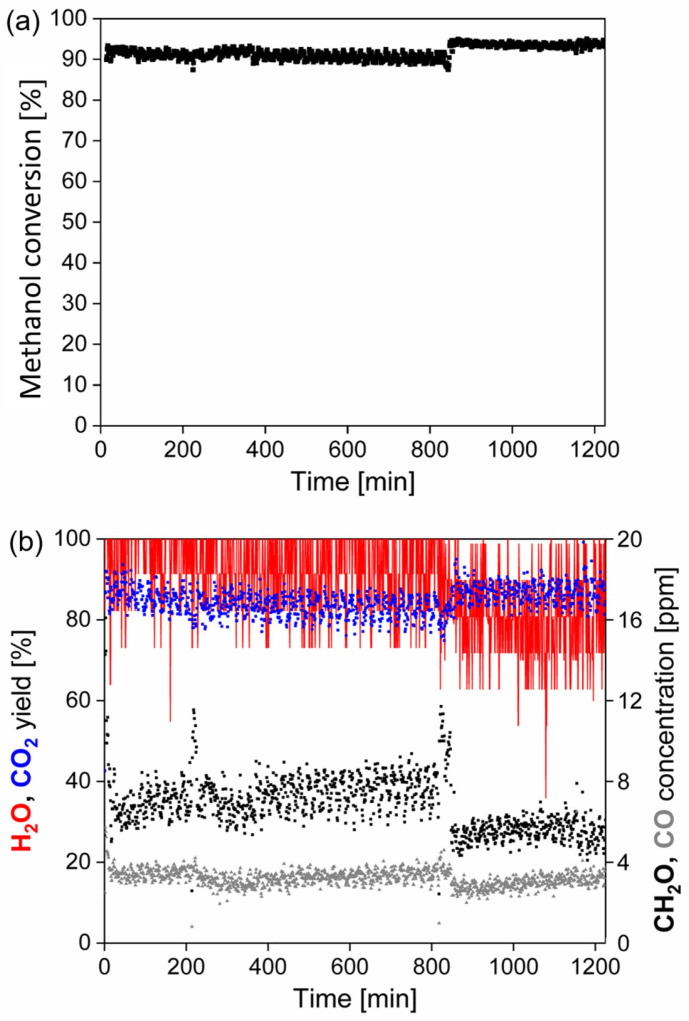
20 h stability test of the TiO_2_-CuO@VUKOPOR^®^A foam in methanol oxidation at T_90_ ~233 °C: (**a**) methanol conversion along reaction time and (**b**) yields of main reaction products (CO_2,_ H_2_O, in %) and concentration of unwanted reaction by-products (CH_2_O, CO, in ppm) along reaction time. Red line—H_2_O yield (in %), blue dots—CO_2_ yield (in %), black dots—detected CH_2_O concentration (in ppm), grey dots—detected CO concentration (in ppm).

**Table 2 nanomaterials-13-01148-t002:** Chemical composition of active phases of the investigated TiO_2_, CeO_2_ and CuO-supported VUKOPOR^®^A ceramic foams determined by XRF.

Catalyst	(wt.%)			(mol.%)		
	TiO_2_	CeO_2_	CuO	Ti	Ce	Cu
TiO_2_-CeO_2_-CuO foam	86	11	3	90	7	3
TiO_2_-CeO_2_ foam	93	7	0	97	3	0
TiO_2_-CuO foam	97	0	3	97	0	3

**Table 3 nanomaterials-13-01148-t003:** Textural properties of the investigated TiO_2_, CeO_2_ and CuO-supported VUKOPOR^®^A ceramic foams determined by nitrogen physisorption at 77 K.

Catalyst	S_BET_ (m^2^ g^−1^)	V_total_ (mm^3^_liq_ g^−1^)
TiO_2_-CeO_2_-CuO foam	59	126
TiO_2_-CeO_2_ foam	58	129
TiO_2_-CuO foam	37	119

**Table 4 nanomaterials-13-01148-t004:** Structural properties of active phases of the investigated TiO_2_, CeO_2_ and CuO-supported VUKOPOR^®^A ceramic foams determined by XRD.

Catalyst	Phase Composition/Crystallite-Size (nm)		
	TiO_2_	CeO_2_	CuO
TiO_2_-CeO_2_-CuO foam	Anatase/12	Cerianite/nuclei < 4 nm	n.d.
TiO_2_-CeO_2_ foam	Anatase/9	Cerianite	n.d.
TiO_2_-CuO foam	Anatase/33 Rutile	n.d.	Tenorite

n.d.—not determined.

**Table 5 nanomaterials-13-01148-t005:** Reducibility, acidity and basicity of the investigated TiO_2_, CeO_2_ and CuO-supported VUKOPOR^®^A ceramic foams determined by H_2_-TPR, NH_3_-TPD and CO_2_-TPD.

Catalyst	Reducibility		Acidity	Basicity
	H_2_-TPR ^a^ (mmol g^−1^)	H_2_-TPR ^b^ (mmol g^−1^)	NH_3_-TPD ^a^ (mmol g^−1^)	CO_2_-TPD ^a^ (mmol g^−1^)
TiO_2_-CeO_2_-CuO foam	1.07	0.42	0.36	0.23
TiO_2_-CeO_2_ foam	0.54	0.94	0.27	0.18
TiO_2_-CuO foam	0.71	0.40	0.20	0.20

^a^ 0–500 °C, ^b^ 500–800 °C.

**Table 6 nanomaterials-13-01148-t006:** Catalytic activity and selectivity of the investigated TiO_2_, CeO_2_ and CuO-supported VUKOPOR^®^A ceramic foams in methanol oxidation.

Catalyst	T_50_ (°C)	T_90_ (°C)	T_95_ (°C)	C_max_ (%)	Main Products		By-Products *	
					Y_CO2,max_ (%)	Y_H2O,max_ (%)	CH_2_O (ppm)	CO (ppm)
TiO_2_-CeO_2_-CuO foam	203	237	253	98	94	86	13	9
TiO_2_-CeO_2_ foam	231	267	279	100	25	74	125	393
TiO_2_-CuO foam	205	233	246	97	100	90	20	2

***** Maximum detected concentration, C_max_ (%)—maximum achieved conversion of methanol, Y_CO2,max_—CO_2_ yield at maximum methanol conversion, Y_H2O,max_—water yield at maximum methanol conversion, T_50_ (°C)—temperature of 50% conversion of methanol, T_90_ (°C)—temperature of 90% conversion of methanol, T_95_ (°C)—temperature of 95% conversion of methanol.

**Table 7 nanomaterials-13-01148-t007:** Catalytic activity and selectivity of the investigated TiO_2_, CeO_2_ and CuO-supported VUKOPOR^®^A ceramic foams in dichloromethane oxidation.

Catalyst	T_50_ (°C)	T_80_ (°C)	C_max_ (%)	Main Products			By-Products *		
				c_max_(CO_2_) (ppm)	Y_CO2,max_ (%)	Y_HCl,max_ (%)	CH_2_O (ppm)	CO (ppm)	CHCl_3_ (ppm)
TiO_2_-CeO_2_-CuO foam	421	497	82	465	91	76	0	15	6
TiO_2_-CeO_2_ foam	377	455	89	128	25	62	12	305	0
TiO_2_-CuO foam	421	492	83	426	81	66	0	2	9

* Maximum detected concentration, T_50_ (°C)—temperature of 50% dichloromethane conversion, T_80_ (°C)—temperature of 80% dichloromethane conversion, C_max_ (%)—maximum achieved dichloromethane conversion, c_max_ (CO_2_) (v ppm)—maximum achieved concentration of CO_2_, Y_CO2,max_—CO_2_ yield at maximum dichloromethane conversion, Y_HCl,max_—yield of HCl at maximum dichloromethane conversion.

**Table 8 nanomaterials-13-01148-t008:** Cl determination in the investigated TiO_2_, CeO_2_ and CuO-supported VUKOPOR^®^A ceramic foams after dichloromethane oxidation.

Catalyst	DisMC	EDX-SEM
	Cl * (µg_Cl_ g^−1^_cat_)	Cl (wt.%)
Parent foam	1.4	n.d.
TiO_2_-CeO_2_-CuO foam	0.7	0.43
TiO_2_-CeO_2_ foam	20.4	0.72
TiO_2_-CuO foam	9.5	1.12

* Experimental error is ±0.7 µg_Cl_ g^−1^_cat_.

## Data Availability

The data presented in this study are available on request from the corresponding author. All results have been published in this article and the original data (notebooks, raw data files from experiments and characterization) have been self-archived according to the instructions given by IET, CEET, VSB-Technical University of Ostrava.

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
