# Peer review of "Oxidation of Methanol and Dichloromethane on TiO_2_-CeO_2_-CuO, TiO_2_-CeO_2_ and TiO_2_-CuO@VUKOPOR^®^A Ceramic Foams"

_nanomaterials, 2023, doi:10.3390/nano13071148_

Round 1

Reviewer 1 Report

Report on the submitted research article nanomaterials-2262501

 In this work the authors investigate the catalytic activity and selectivity of three different types of mixed oxide catalysts. More specifically TiO2, CeO2 and CuO-based open-cell foam supported catalysts, were tested for fist time regards to the methanol and dichloromethane oxidation. All the catalysts were synthesized with the sol-gel method and supported on porous VUKOPOR®A ceramic foams. A big variety of techniques were used in order to characterize the supported catalysts from structural, morphological, physicochemical and catalytic point of view. The results demonstrate that the TiO2-CuO@VUKOPOR®A foam showed the best catalytic activity and CO2 yield in methanol oxidation with a good catalytic stability. On the other hand, the TiO2-CeO2-CuO@VUKOPOR®A foam was the best catalyst in dichloromethane oxidation. A series of reasons are given by the authors for this catalytic behavior, based on the reducibilty, acidity and basicity of the materials.

            In my opinion this is an interesting work because it deals with the catalytic decomposition of two harmful and toxic solvents such as the methanol and dichloromethane are. It also concerns the reduction of their volatile emissions. In general, the whole work looks quite comprehensive. The manuscript is well written, the experimental methods are sufficiently described and the results and conclusions are clearly shown. The references are adequate. On balance the article is suitable for publication in Nanomaterials. However, before the final publication, the authors should pay attention on the following major and minor comments and revise the manuscript accordingly.  

Majors

1)        Table 2: For the TiO2-CeO2-CuO foam if we add all the mol% concentrations (91+7+3) we get 101%. This is not correct.

2)        Page 9: Crystal structure and crystallinity. The authors should give some information how the crystallites sizes (see Table 4) were calculated.

3)        Figure 4: In XRD patterns is the VUKOPOR®A foam involved or it is only the oxides’ powders? In the first case the VUKOPOR®A ceramic foam should had given some serious traces of alumina and silica too.

4)        Lines 313-314: How the mass of each deposited catalyst was measured? Does this mass include the ceramic foam VUKOPOR®A foam? The same question in lines 344-345.

5)        Equation 5: Is the first part of the equation YHCL max or simply YHCL? Also in the denominator, why the factor 2 is used? The same question holds for equation 3.

6)        Figure 5: The authors define TPR peaks and shoulders with an accuracy of 1oC. In some cases however the peaks are not clearly defined (see for example the TiO2-CeO2-CuO and TiO2-CuO cases especially within the region 500-800 oC). What is the real experimental accuracy of the temperature and how reliable are the mentioned temperatures in the figure?

7)        Table 5: How all these numbers were calculated? Are they representing the areas under the recorded curves in Figures 5,6 and 7? If yes this should be pointed out in the text.

8)        Line 557: Do the authors consider the TiO2-CeO2 foam less active because of the higher T50, T90 and T95 temperatures compared to those of the two other foams. If yes, this criterion should be pointed out in the text.

9)        Figure 9b: The dotted curve goes higher than 100%. The y axis is not correct.

10)    Lines 815-821: It is not clear to me how Cl presence is evident looking at the SEM images. Could the authors be more explanatory? Also in Fig.15, it should be mentioned separately what each color means at the different images.

Minors

1)      Line 75: Ref 12 does not look suitable for here. It does not mention titania at all.

2)      Line 77: Ref 3 looks to me irrelevant. Please verify.

3)      Line 247: What TCD is? The authors should explain this abbreviation.

4)      Line 446: Table 5 instead of table 4.

5)      Fig. 10: An inset label in each figure describing the foam which relates will be more convenient to the reader. Also, in fig. 10c, I see 20% ppm for CH2O conversion instead of 19% written in the text (lines 565 and 653). What is the right value?

6)      Figure 14: I think T80=455 oC instead of 452 (see table 7).

7)      Table 8: I guess the first column is referred to DisMC measurements. Looking at the Table 8 and its caption, this is not clear.

Author Response

Majors

1)        Table 2: For the TiO2-CeO2-CuO foam if we add all the mol% concentrations (91+7+3) we get 101%. This is not correct.

Response: We apologize for this numerical mistake. It was corrected in Table 2 for TiO2-CeO2-CuO catalytic foam to 90+7+3 mol.%.

2)        Page 9: Crystal structure and crystallinity. The authors should give some information how the crystallites sizes (see Table 4) were calculated.

Response: The following information was added to the manuscript to the section 2.2. Physicochemical characterization of TiO2, CeO2 and CuO-supported VUKOPOR®A  ceramic foams:

´Phase composition was determined using the reference diffractograms database ICDD (PDF-2). The size of the crystallites was calculated based on the Scherrer equation with a diffractometer resolution correction to LaB6 standard.´

3)        Figure 4: In XRD patterns is the VUKOPOR®A foam involved or it is only the oxides’ powders? In the first case the VUKOPOR®A ceramic foam should had given some serious traces of alumina and silica too.

Response: In Figure 4 only the oxides’ powders are displayed. As it is written in the section 2.2. Physicochemical characterization of TiO2, CeO2 and CuO-supported VUKOPOR®A  ceramic foams, the powder equivalents of the catalysts deposited (i.e. oxides´ powders without foam) were analyzed using XRD. Therefore, only TiO2 anatase and rutile and CuO tenorite are evident in XRD diffractograms, without any evidence of Al2O3 or SiO2 phases.

4)        Lines 313-314: How the mass of each deposited catalyst was measured? Does this mass include the ceramic foam VUKOPOR®A foam? The same question in lines 344-345.

Response: We have to confess that the knowledge of the each mass of each catalyst deposited on the each VUKOPOR®A foam was very meticulous issue during their preparation. But, it was very important for us to know the catalyst masses for chemical, textural, TPR and TPD characterizations as well as catalytic testing. The mentioned individual masses in the manuscript are the masses only of the deposited oxides, without the VUKOPOR®A ceramic foam. VUKOPOR®A ceramic foam itself is nonporous and does not show any catalytic activity.

Overall, for each catalyst (i.e. TiO2-CeO2-CuO, TiO2-CeO2, TiO2-CuO), 60 pieces of catalyst@VUKOPOR®A foam were prepared. Before deposition, each of 60 pieces of VUKOPOR®A foam was weighted, then the catalyst was repeatedly deposited from the sol by dip-coating and calcined (4 times ~ 4 layers were deposited) and then the final oxide catalyst weight deposited on each VUKOPOR®A foam was determined. During the preparation, none of the foams should be interchanged in order to know the final catalyst mass on each piece of VUKOPOR®A foam.

5)        Equation 5: Is the first part of the equation YHCL max or simply YHCL? Also in the denominator, why the factor 2 is used? The same question holds for equation 3.

Response: Equation (5) describes the calculation of HCl yield (in %). Thus, it should be YHCl, but the text was corrected accordingally as follows:

The HCl yield, YHCl, was calculated according to Eq. (5):

                                (5) (visible in attached file)

where YHCl is the HCl yield (%), cHCl is the measured output concentration of HCl (volume ppm) and cDCM,0 is the initial dichloromethane concentration (volume ppm).

Due to the FTIR analysis, the complete closing of the chlorine-balance is not possible, since the formation of Cl2 was not measured. The selectivity towards HCl, the desired chlorinated oxidation product, was followed by measuring the HCl concentration at the outlet and the HCl yield was calculated according to Equation (5). The factor 2 is used because of dichloromethane (CH2Cl2) should theoretically yield HCl in concentration equals 2 x cDCM,0. For example, with 500 ppm of dichloromethane in the feed the 100% yield of HCl would be 1000 ppm.

CH2Cl2 + O2  = 2 HCl + CO2                                                                            

Concerning Equation (3), the factor 2 was used since methanol (CH3OH) should theoretically yield H2O in concentration equals 2 x cMeOH,0. For example, with 500 ppm of methanol in the feed the 100% yield of H2O would be 1000 ppm.

CH3OH + 3/2 O2 = CO2 + 2 H2O                                                                          

6)        Figure 5: The authors define TPR peaks and shoulders with an accuracy of 1oC. In some cases however the peaks are not clearly defined (see for example the TiO2-CeO2-CuO and TiO2-CuO cases especially within the region 500-800 oC). What is the real experimental accuracy of the temperature and how reliable are the mentioned temperatures in the figure?

Response: For all measured temperature-programmed techniques (NH3-TPD, CO2-TPD and H2-TPR) the experimental error of measurements of temperature expressed as a relative standard deviation, determined on the basis of repeated tests of standards (Co3O4 and CoO), is 1.4 %. This information was added to the section 2 Material and methods, 2.2. Physicochemical characterization of TiO2, CeO2 and CuO-supported VUKOPOR®A ceramic foams.

7)        Table 5: How all these numbers were calculated? Are they representing the areas under the recorded curves in Figures 5,6 and 7? If yes this should be pointed out in the text.

Response: For all measured temperature-programmed spectra (NH3-TPD, CO2-TPD and H2-TPR) the area of the peak was evaluated as the area under the curve in the region directly defined by the temperature without any deconvolution procedure. This information was added to the section 2 Material and methods, 2.2. Physicochemical characterization of TiO2, CeO2 and CuO-supported VUKOPOR®A  ceramic foams.

8)        Line 557: Do the authors consider the TiO2-CeO2 foam less active because of the higher T50, T90 and T95 temperatures compared to those of the two other foams. If yes, this criterion should be pointed out in the text.

Response: In chapter 2.3. TiO2, CeO2 and CuO-supported VUKOPOR®A ceramic foams testing in methanol and dichloromethane oxidation there is stated:

´Temperatures T50, T90, T95 and Cmax corresponding to 50, 90, 95% and the maximum achieved methanol conversion were chosen as comparative measures of the catalyst activity.´

As required, this information was repeated also in the chapter 3.2 Catalytic results of TiO2, CeO2 and CuO-supported VUKOPOR®A ceramic foams in the oxidation of methanol and dichloromethane.

9)        Figure 9b: The dotted curve goes higher than 100%. The y axis is not correct.

Response: We realize that CO2 yield (in %) (Figure 9b) is at some temperatures higher than 100%. This feature arises from the fact that some adsorption of methanol occurrs on the catalytic foams at lower temperatures. This is reflected at higher temperatures in higher CO2 yield due to proceeding methanol desorption and its oxidation. The adsorption of methanol is visible from the light-off curves of methanol at temperature 50-70°C (see Figure 9a).

10)    Lines 815-821: It is not clear to me how Cl presence is evident looking at the SEM images. Could the authors be more explanatory? Also in Fig.15, it should be mentioned separately what each color means at the different images.

Response: Figure 15 was corrected, increasing the photos labels being better readable. Each color represents different element mapping detected over the material photo, which is now better evident. We very apologize for original poor readability of the photos.

The main point of photos is to show where Al, O, Ti, Ce, Cu and Cl (main elements of catalytic foams) are present/located along the designed catalytic foams. While Al is representing the VUKOPOR®A ceramic foam, Ti, Ce, Cu and Cl represents the location of deposited oxide catalyst. O may represent both. It is evident that Ti and Ce are well-visible and distributed in the similar locations in TiO2-CeO2-CuO@VUKOPOR®A and TiO2-CeO2@VUKOPOR®A foams.  However, Cu in TiO2-CeO2-CuO@VUKOPOR®A and TiO2-CuO@VUKOPOR®A and Cl for all three catalytic foams are poorly-visible because of their low concentrations. But still, there you can see the detected color dot-points of Cu (green point) and Cl (pink points) in the photos. For TiO2-CeO2@VUKOPOR®A foam (Figure 15b) Ti and Ce are regularly distributed and is not clear where Cl is prefferentially located. For TiO2-CuO@VUKOPOR®A foam (Figure 15c) Cl is on different places than Cu, Cl is in the contact with Ti locations. For TiO2-CeO2-CuO@VUKOPOR®A foam (Figure 15a) Ce is located on some places with Ti, but also separately from Ti. Cu is located close to places with Ce. Cl is practically not visibible/was poorly detected.

Minors

1)      Line 75: Ref 12 does not look suitable for here. It does not mention titania at all.

Response: Thank you very much for this remark. It was a mistake. We very apologize. It was corrected.

2)      Line 77: Ref 3 looks to me irrelevant. Please verify.

Response: Thank you very much for this remark. It is the truth that the Ref 3 is inrrelevant on line 77, it was a mistake. It was corrected.

3)      Line 247: What TCD is? The authors should explain this abbreviation.

Response: TCD is a ´thermal conductivity detector´. It was added to the manuscript.

4)      Line 446: Table 5 instead of table 4.

Response: Thank you very much. It was corrected.

5)      Fig. 10: An inset label in each figure describing the foam which relates will be more convenient to the reader. Also, in fig. 10c, I see 20% ppm for CH2O conversion instead of 19% written in the text (lines 565 and 653). What is the right value?

Response: We thank the reviewer for this remark and recommendation. We considered the addition of inset label describing the type of foam, but finally we suppose since Figure 10, Figure 12 and also Figure 15 are very similar concerning their figure captions, we would like to leave all figures as they are.

Concerning the ppm of CH2O, the right value is 20 ppm. It was corrected in the Table 6 and proper lines in the text.

6)      Figure 14: I think T80=455 oC instead of 452 (see table 7).

Response: Yes, in Figure 14 there should be 455°C. It was corrected.

7)      Table 8: I guess the first column is referred to DisMC measurements. Looking at the Table 8 and its caption, this is not clear.

Response: The information about DisMC and EDX-SEM analyses was added directly to Table 8.

Reviewer 2 Report

Catalytic oxidation of VOC is a practically important topic of modern catalysis. The influence of the methods and conditions of oxide-based catalysts preparation on features and behavior of systems.

In the present manuscript detailed study of TiO2, CeO2 and CuO – supported VUKOPORA ceramic foams, prepared using a reverse micelles-controlled sol-gel method, in methanol oxidation and in dichloromethane oxidation.

The minor comments:

1) Abstract:

 These solvents are widely used in pharma-18 ceutical and chemical industry and their gaseous emissions must be reduced effectively because of 19 toxic and harmful effects.

This sentence is not wanted here.

2) Introduction

Concerning catalytic oxidation of a broad range of VOCs over individual types 61 of catalysts some review papers were already reported [1,2,4-6], also including investi-62 gated confined-structured catalysts [6] and by-products analysis for chlorinated VOCs 63 (CVOCs) oxidation [7].

So what interesting or important for the present study was reported in these reviews?

These references are not discussed further.

3) Figure 15: Figure is unreadable. Zooming does not help.

Major comment:

Authors emphasized that the application of the appropriate method of preparation allowed to get presented results in catalysis. However, there is no discussion of the mechanisms of the formation of the systems during preparation under used conditions.

Thus for the dichloromethane oxidation presence of Cu – component is needed for the CO2 selectivity but not good for CHCl3 (since CuO bulk phase is responsible for that). At the same time, the method allows formation of the dispersed bulk phase CuO, which looks promising. So does the used method of preparation allow optimization of the systems composition? Is it possible that the Cu / Ce / Ti ratio at the stage of preparation can be optimized for better results in catalytic reactions? Did authors try to do that?

So the mechanisms of system formation during preparation should be discussed from the standpoint of the influence on the catalytic behavior of catalysts.

Manuscript can be published, but mentioned comments should be discussed. 

Author Response

The minor comments:

1) Abstract:

These solvents are widely used in pharma-18 ceutical and chemical industry and their gaseous emissions must be reduced effectively because of 19 toxic and harmful effects.

This sentence is not wanted here.

Response: The sentence was removed from the abstract.

2) Introduction

Concerning catalytic oxidation of a broad range of VOCs over individual types 61 of catalysts some review papers were already reported [1,2,4-6], also including investi-62 gated confined-structured catalysts [6] and by-products analysis for chlorinated VOCs 63 (CVOCs) oxidation [7].

So what interesting or important for the present study was reported in these reviews?

These references are not discussed further.

Response: The following text was added to the Introduction part of the manuscript, summarizing the main challenges defined within the reported reviews.:

´The main challenges which need to be still further solved concerning the development of catalysts for VOCs oxidation are the development of highly selective and durable catalysts, deeper knowledge and prediction of oxidation mechanisms [4,5], catalyst deactivation, development of more complex catalytic systems being able to oxidize a range of VOCs at much lower temperatures [1] and with respect to character of contaminated sites the coupling process combining different VOCs/CVOCs abatement methods should be applied to enhance the removal efficiency in a more cost effective way [1,2,5]. Concerning the confined-structured catalysts more easy, available and green synthetic methods should be developed and used, including the optimization of chemical composition. There also arise some limitations in their precise physicochemical characterization, which is necessary for understanding of their catalytic performance. There is also a lack of testing of confined-structured catalysts in more realistic working conditions and oxidation of a mixture of VOCs [6]. For CVOCs oxidation, (i) the examination of Cl migration and evolution pathway via advanced modelling and characterization techniques, providing the information how to prevent unwanted chlorinated by-products formation with working adjustment, (ii) revelation of proper metal combination within catalyst design, (iii) definition of the exact role of each physicochemical property in deep oxidation and chlorination steps with the aim to tailor catalyst surface composition, inhibiting the formation of unwanted chlorinated by-products and (iv) enhancing oxidation efficiency at low temperature that can evade Cl2 formation and chlorination reactions, are of keen interests. It was suggested that e.g. technological coupling of hydrodechlorination, catalytic oxidation and ozone-assisted oxidation is feasible approach how to enhance total elimination at low temperature without chlorination reactions. (v) Last but not least, investigation of the effect of other additives and oxidizing agents, promoting dechlorination reactions and inducing Cl species into final products, is also necessary [7]. Based on above mentioned facts the development of industrially-attractive and robust form of catalyst based on mixed metal oxides, which are cheaper and more resistant against metal chlorination than noble metal-based catalysts, using simple chemical and deposition route, is the logical motivation for the research. Moreover, the developed form of catalysts should be examined in oxidation of more model VOCs and in the presence of water to reveal the catalyst selectivity and be closer to realistic adjustment, yielding more desired HCl than Cl2

3) Figure 15: Figure is unreadable. Zooming does not help.

Response: Figure 15 was corrected to be readable and replaced the original one.

Major comment:

Authors emphasized that the application of the appropriate method of preparation allowed to get presented results in catalysis. However, there is no discussion of the mechanisms of the formation of the systems during preparation under used conditions.

Thus for the dichloromethane oxidation presence of Cu – component is needed for the CO2 selectivity but not good for CHCl3 (since CuO bulk phase is responsible for that). At the same time, the method allows formation of the dispersed bulk phase CuO, which looks promising. So does the used method of preparation allow optimization of the systems composition? Is it possible that the Cu / Ce / Ti ratio at the stage of preparation can be optimized for better results in catalytic reactions? Did authors try to do that?

So the mechanisms of system formation during preparation should be discussed from the standpoint of the influence on the catalytic behavior of catalysts.

Response: The used reverse micelles-assisted sol-gel method enables to tune the molar composition of Ti:Ce:Cu. It is the truth that the different Ti:Ce:Cu molar composition can result in the catalyst with different structural properties. However, within our manuscript we did not study the effect of Ti:Ce:Cu molar composition on the physicochemical properties of the catalytic foams and subsequently on the catalytic performance. The Ti:Ce:Cu molar composition in our study was designed according to the reported study of Cao et al. [25] who studied a two-step catalytic system Ce/TiO2–Cu/CeO2 in a series.

Within our research we are currently working on the investigation of the effect of preparation procedure of the catalyst of the same Ti:Ce:Cu molar composition on its physicochemical properties and catalytic efficiency in dichloromethane and methanol oxidation. The preparation procedure has the significant effect on the micro/structure of the catalyst, thus, we assume that it will be reflected to its catalytic performance. The second interesting issue under our keen interest is the attachment and the mechanical stability of the catalyst layer deposited on the VUKOPOR@A ceramic foam using different preparation procedures. It is clear that the suspension deposition offers less mechanically-stable catalytic layers on the ceramic foams than sol-gel method when nanoparticles crystalize during subsequent calcination in a thinner layer.

[25] Cao, S.; Shi, M.P.; Wang, H.Q.; Yu, F.X.; Weng, X.L.; Liu, Y.; Wu, Z.B. A two-stage Ce/TiO2-Cu/CeO2 catalyst with separated catalytic functions for deep catalytic combustion of CH2Cl2. Chem Eng J 2016, 290, 147-153, doi:10.1016/j.cej.2015.12.107.

Manuscript can be published, but mentioned comments should be discussed. 

Reviewer 3 Report

This paper describes an original and interesting work. As such, it has the potential to be published in Nanomaterials. However, I have the following comments that the authors should carefully implement in the revised manuscript prior to publication.

1) In order to give a more complete picture, the authors should point out, citing relevant literature, that nano ceria and even more Cu-doped nano ceria are also very effective catalysts in oxidation reactions other than those involving VOCs and, more specifically, in the oxidation of particulate matter (PM) (i.e., regeneration of particulate filters). In this regard, the following works should be cited: AIChE Journal, 2017, 63(8), pp. 3442–3449; Topics in Catalysis, 2021, 64(3-4), pp. 256–269.

2) Introduction - The connection between the aim of the work and the literature gaps should be better described, thus giving more strength to the reason for this work.

3) Results and discussion/Conclusions - The practical impact of the results obtained in this work should be better highlighted.

4) Conclusions - The authors should also give an outlook on future research work.

I’m willing to review the revised manuscript.

Author Response

1) In order to give a more complete picture, the authors should point out, citing relevant literature, that nano ceria and even more Cu-doped nano ceria are also very effective catalysts in oxidation reactions other than those involving VOCs and, more specifically, in the oxidation of particulate matter (PM) (i.e., regeneration of particulate filters). In this regard, the following works should be cited: AIChE Journal, 2017, 63(8), pp. 3442–3449; Topics in Catalysis, 2021, 64(3-4), pp. 256–269.

Response: The mentioned fact was added and literature was cited in the Introduction part of manuscript.

2) Introduction - The connection between the aim of the work and the literature gaps should be better described, thus giving more strength to the reason for this work.

Response: The following text was added to the Introduction part of the manuscript, summarizing the main challenges defined within the reported reviews. The reason of the work was also pointed out.

´The main challenges which need to be still further solved concerning the development of catalysts for VOCs oxidation are the development of highly selective and durable catalysts, deeper knowledge and prediction of oxidation mechanisms [4,5], catalyst deactivation, development of more complex catalytic systems being able to oxidize a range of VOCs at much lower temperatures [1] and with respect to character of contaminated sites the coupling process combining different VOCs/CVOCs abatement methods should be applied to enhance the removal efficiency in a more cost effective way [1,2,5]. Concerning the confined-structured catalysts more easy, available and green synthetic methods should be developed and used, including the optimization of chemical composition. There also arise some limitations in their precise physicochemical characterization, which is necessary for understanding of their catalytic performance. There is also a lack of testing of confined-structured catalysts in more realistic working conditions and oxidation of a mixture of VOCs [6]. For CVOCs oxidation, (i) the examination of Cl migration and evolution pathway via advanced modelling and characterization techniques, providing the information how to prevent unwanted chlorinated by-products formation with working adjustment, (ii) revelation of proper metal combination within catalyst design, (iii) definition of the exact role of each physicochemical property in deep oxidation and chlorination steps with the aim to tailor catalyst surface composition, inhibiting the formation of unwanted chlorinated by-products and (iv) enhancing oxidation efficiency at low temperature that can evade Cl2 formation and chlorination reactions, are of keen interests. It was suggested that e.g. technological coupling of hydrodechlorination, catalytic oxidation and ozone-assisted oxidation is feasible approach how to enhance total elimination at low temperature without chlorination reactions. (v) Last but not least, investigation of the effect of other additives and oxidizing agents, promoting dechlorination reactions and inducing Cl species into final products, is also necessary [7]. Based on above mentioned facts the development of industrially-attractive and robust form of catalyst based on mixed metal oxides, which are cheaper and more resistant against metal chlorination than noble metal-based catalysts, using simple chemical and deposition route, is the logical motivation for the research. Moreover, the developed form of catalysts should be examined in oxidation of more model VOCs and in the presence of water to reveal the catalyst selectivity and be closer to realistic adjustment, yielding more desired HCl than Cl2

3) Results and discussion/Conclusions - The practical impact of the results obtained in this work should be better highlighted.

Response: The following statement was added to the Conclusions:

´Findings presented within this study indicate a good potential of used sol-gel route for preparation of deposited thin film catalysts on the ceramic types of supports suitable for catalytic oxidation of VOCs in industrial scale.´

4) Conclusions - The authors should also give an outlook on future research work.

Response: The following statement was added to the Conclusions:

´Future work should be focused on the study of the effect of chemical composition (Ti:Ce:Cu molar ratio) of the deposited catalyst on its catalytic performance in individual VOCs oxidation. From more general point of view, the catalyst design and VOCs catalytic oxidation should follow the development direction that unwanted oxidation by-products are minimized and simultaneously NOx is not produced. These are the reasons why catalysts needs further research and each VOC emissions needs to be looked at.´

Round 2

Reviewer 1 Report

The revised manuscript is ready for publication.

Reviewer 2 Report

Manuscript can be published

Reviewer 3 Report

The authors have addressed my comments in a satisfactory manner. Overall, the manuscript has been improved after revisions. Therefore, it can be accepted for publication in Nanomaterials.